



# MAX-DOAS measurements of tropospheric NO$_2$ and HCHO in Munich and the comparison to OMI and TROPOMI satellite observations

Ka Lok Chan[1], Matthias Wiegner[2], Carlos Alberti[2,3], and Mark Wenig[2]

[1]Remote Sensing Technology Institute (IMF), German Aerospace Center (DLR), Oberpfaffenhofen, Germany
[2]Meteorological Institute (MIM), Ludwig Maximilians Universität München (LMU), Munich, Germany
[3]Institute of Meteorology and Climate Research (IMK-ASF), Karlsruhe Institute of Technology (KIT), Karlsruhe, Germany

**Correspondence:** Ka Lok Chan (ka.chan@dlr.de)

**Abstract.**

We present two dimensionally scanning Multi-AXis Differential Optical Absorption Spectroscopy (MAX-DOAS) observations of nitrogen dioxide (NO$_2$) and formaldehyde (HCHO) in Munich. Vertical columns and vertical distribution profiles of aerosol extinction coefficient, NO$_2$ and HCHO are retrieved from the 2D MAX-DOAS observations. The measured surface aerosol extinction coefficients and NO$_2$ mixing ratios derived from the retrieved profiles are compared to in-situ monitor data, and the surface NO$_2$ mixing ratios show good agreement with in-situ monitor data with a Pearson correlation coefficient ($R$) of 0.91. The aerosols optical depths (AODs) show good agreement as well ($R = 0.80$) when compared to sun-photometer measurements. Tropospheric vertical column densities (VCDs) of NO$_2$ and HCHO derived from the MAX-DOAS measurements are also used to validate OMI and TROPOMI satellite observations. Monthly averaged data show good correlation, however, satellite observations are on average 30 % lower than the MAX-DOAS measurements. Furthermore, the MAX-DOAS observations are used to investigate the spatio-temporal characteristic of NO$_2$ and HCHO in Munich. Analysis of the relations among aerosol, NO$_2$ and HCHO shows higher aerosol to HCHO ratios in winter indicating a longer atmospheric lifetime of aerosol and HCHO. The analysis also suggests that secondary aerosol formation is the major source of aerosols in Munich.

## 1 Introduction

Nitrogen dioxide (NO$_2$) and formaldehyde (HCHO) are important atmospheric constituents that can have a strong influence on air quality and climate. Both play a crucial role in the formation of tropospheric ozone (O$_3$) (Crutzen, 1970) and aerosols (Jang and Kamens, 2001), consequently having a strong impact on the Earth's radiation budget. Moreover, they are toxic to humans in high concentrations. Major sources of NO$_2$ are fossil fuel combustion, biomass burning, lightning and oxidation of ammonia (Bond et al., 2001; Zhang et al., 2003). HCHO is an intermediate product of the oxidation of almost all volatile organic compounds (VOCs), which is why it is widely used as an indicator of non-methane volatile organic compounds (NMVOCs) (Fried et al., 2011). VOCs also have a significant impact on the atmospheric abundance of hydroxyl (OH) radicals, which are the major oxidants in the troposphere. The main HCHO sources include oxidation of VOCs emitted from plants, biomass





burning, traffic and industrial emissions. Despite the importance of HCHO, it is typically not considered a gas that has to be regularly monitored, so more measurements are needed in order to fully examine atmospheric processes involving HCHO.

Space-borne observations are indispensable tools to monitor the spatio-temporal distribution of atmospheric pollutants like $NO_2$ and HCHO on a global scale (Burrows et al., 1999; Bovensmann et al., 1999; Callies et al., 2000; Levelt et al., 2006; Veefkind et al., 2012). Vertical Column Densities (VCDs), representing concentrations integrated over vertical atmospheric columns, derived from spectral radiances provide deeper insights into atmospheric dynamics, as well as anthropogenic and natural emissions (Beirle et al., 2003; Wenig et al., 2003; Beirle et al., 2004; Richter et al., 2005; Zhang et al., 2007; van der A et al., 2008). However, the accuracy of satellite retrievals strongly depends on a number of assumptions about the surface albedo, cloud and aerosol optical properties, and the vertical distribution of trace gases. Therefore, validation of satellite obser-vations by means of ground-based observations is crucial to determine the influence of those assumptions on the accuracy of the VCDs (Wenig et al., 2008; Chen et al., 2009; Lin et al., 2014; Chan et al., 2015; Jin et al., 2016). Furthermore, the temporal sampling of satellite measurements is typically limited to a small number of overpasses per day prohibiting observation of diurnal cycles. In order to derive a complete picture of spatio-temporal variability, the combination of space-borne and ground based observations are useful.

The Multi-AXis Differential Optical Absorption Spectroscopy (MAX-DOAS) technique measures the vertical distribution of $NO_2$, HCHO and aerosols. This passive remote sensing technique uses spectroscopic observations of scattered sun-light under different viewing directions and the differential optical absorption spectroscopy (DOAS) technique (Platt and Stutz, 2008) to derive column densities from molecular absorption in ultraviolet and visible spectral bands. Because of its compact experimental setup, it has been widely used for ground based observations (Hönninger and Platt, 2002; Hönninger et al., 2004; Wittrock et al., 2004; Frieß et al., 2006; Irie et al., 2008; Li et al., 2010; Clémer et al., 2010; Halla et al., 2011; Li et al., 2013; Ma et al., 2013; Chan et al., 2015; Wang et al., 2016) as well as for satellite validation (Jin et al., 2016; Chan et al., 2018).

Although the $NO_2$ load in many parts of the world including Germany show decreasing trends, concentrations in many cities in Germany still exceed the World Health Organization (WHO) annual average limit of $40\,\mu g/m^3$. Such exceedances are recorded at about $40\,\%$ of the traffic oriented monitoring stations (UBA, 2019), constituting one of the most severe air pollution problems in Germany. One example of high concentrations of pollutants is Munich, the German city with the highest $NO_2$ value in 2017 and second hightest in 2018 (UBA, 2019). Munich is the third largest city in Germany with a population of around 1.5 million. Traffic and industrial emissions are the major anthropogenic sources of air pollution in Munich.

Ground-based MAX-DOAS measurements are performed since October 2016. The MAX-DOAS experimental setup, the spectral analysis as well as the retrieval of the aerosol extinction coefficients, $NO_2$ and HCHO concentration profiles are described in Section 2. The results of our retrievals for Munich include the spatial distribution of $NO_2$ and HCHO, their weekly pattern, and the interrelationship between aerosols, $NO_2$ and HCHO are presented in Section 3. Comparisons with independent measurements are discussed in Section 4: aerosol extinction and $NO_2$ mixing ratios at the lowest layer of the MAX-DOAS profile are compared to ground-based in-situ data, and aerosol optical depth (AOD) to sun-photometer measurements. Validation of OMI and TROPOMI satellite observations in terms of $NO_2$ and HCHO vertical column densities (VCDs) is subject of Sections 5. Section 6 concludes our study.



## 2 Data and Methods

### 2.1 2D MAX-DOAS measurements

#### 2.1.1 Experimental setup

A 2D MAX-DOAS instrument was set up on the roof of a university building of the Ludwig-Maximilians-Universität München

(48.148°N, 11.573°E) which is about 25 m above ground level (515 m above sea level). The site is located 1.2 km north to the Munich city center. The locations of the MAX-DOAS as well as an in-situ air quality monitoring station in Munich are indicated in Fig. 1. The 2D MAX-DOAS instrument measuring scattered sun-light consists of a scanning telescope, 2 stepping motors controlling the viewing azimuth ($0° \leq \phi \leq 360°$) and elevation angle ($2° \leq \alpha \leq 90°$), and 2 spectrometers covering the ultraviolet (UV) and visible (VIS) wavelength range. Scattered sun-light collected by the telescope is redirected by a prism

reflector and quartz fibers to the spectrometers for spectral analysis. The field of view of the instrument is about 0.4°. Two Avantes AvaBench-75 spectrometers equipped with backthinned Hamamatsu charge-coupled device (CCD) detectors are used to cover UV (305 - 460 nm) and VIS (430 - 650 nm) wavelength ranges, respectively. The full width half maximum (FWHM) spectral resolution of the UV and VIS spectrometer is 0.62 nm and 0.87 nm, respectively.

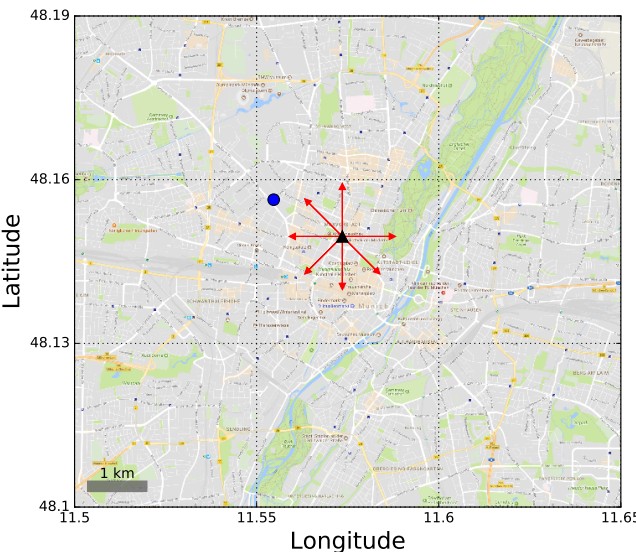

**Figure 1.** Locations of the MAX-DOAS measurement site (black triangle) and the ambient air quality monitoring station (blue dot). The red arrows indicate the azimuth viewing directions $\phi$ of the MAX-DOAS observations. The base map is taken from © Google Maps (https://www.google.com/maps/).

A measurement cycle starts with measuring scattered sun-light spectra at elevation angles ($\alpha$) of 2°, 3°, 4°, 5°, 6°, 8°, 15°,

30° and 90° (zenith) for each azimuth angle ($\phi$). For this study, the MAX-DOAS was configured to measure 7 consecutive





azimuth angles of 0°, 90°, 135°, 180°, 225°, 270° and 315°. Measurements with $\phi = 45°$ were omitted because a building close by is blocking the lower elevation angles. The exposure time and the number of scan of each individual measurement are adjusted automatically depending on the intensity of the received scattered sun-light in order to have similar integration time of 1 minute for all the measurements. A full measurement sequence for all azimuth directions takes about an hour.

5   **2.1.2   Spectral retrieval**

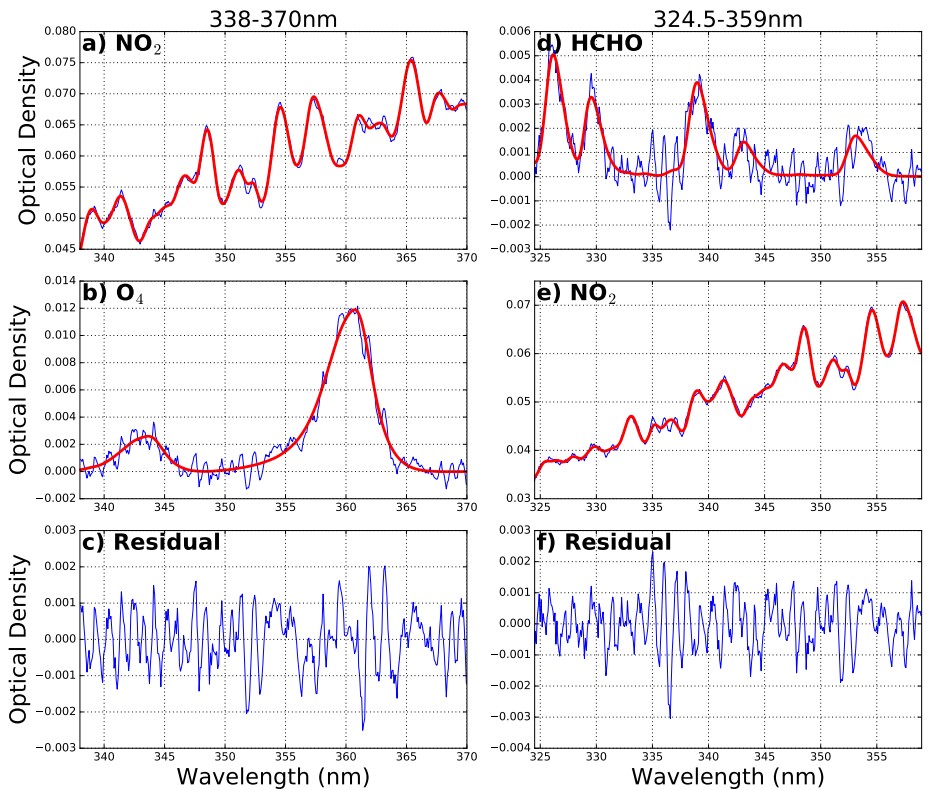

**Figure 2.** An example of the DOAS retrieval of NO$_2$ and HCHO DSCDs from a MAX-DOAS spectrum taken 4 November 2016 at 10:02 (local time) with viewing elevation angle of $\alpha = 2°$. The left panels show the DOAS fit in the wavelength range 338 - 370 nm, while the right panels show DOAS fit in the wavelength range 324.5 - 359 nm.

All measurement spectra were corrected for the spectrometer CCD non-linearity, as well as for offset and dark current. The DOAS technique (Platt and Stutz, 2008) is then applied to the measurement spectra to derive slant column densities (SCDs) of the trace gases. In this study, the measurement spectra are evaluated using the spectral analyzing software QDOAS version 3.2. The spectral fit is performed at two different wavelength bands of 338 - 370 nm and 324.5 - 359 nm. O$_4$ DSCDs used for aerosol

10   extinction profile retrieval and NO$_2$ DSCDs used for NO$_2$ profile retrieval are taken from the former fitting band (338 - 370 nm).





Due to the stronger absorption structure of HCHO in the shorter wavelengths, HCHO DSCDs used for the retrieval of HCHO profiles are taken from the latter fitting window (324.5 - 359 nm). Detailed procedure of the combined retrieval of aerosol and trace gas profiles is presented in Section 2.1.3. The zenith spectrum ($\alpha = 90°$) of the corresponding measurement cycle is used as reference spectrum to retrieve the differential slant column densities (DSCDs), which are defined as the difference between

the SCDs of an off-zenith spectra and the corresponding zenith reference spectrum. A $5^{th}$ order polynomial in the DOAS fit is responsible for removing broadband spectral structures caused by Rayleigh and Mie scattering. The absorption cross section of several trace gases used in the retrieval are listed in Table 1 for both wavelength ranges. These settings are based on the results from previous studies (Pinardi et al., 2013; Peters et al., 2017; Kreher et al., 2019). In order to compensate for possible instabilities due to small thermal variations of the spectrograph, shift and squeeze parameters of the spectra are included in the

fitting process as well. An example of the DOAS retrieval of $NO_2$ and HCHO DSCDs from a MAX-DOAS spectrum taken on 4 November 2016 at 10:02 (local time) with $\alpha = 2°$ is shown in Fig. 2.

**Table 1.** The DOAS retrieval settings for different wavelength bands.

| Species | Temperature | Wavelength Range | | Reference |
| --- | --- | --- | --- | --- |
| | | 324.5 - 359 nm | 338 - 370 nm | |
| BrO | 223 K | ✓ | ✓ | Fleischmann et al. (2003) |
| HCHO | 298 K | ✓ | ✓ | Meller and Moortgat (2000) |
| $NO_2$[a] | 298 K | ✓ | ✓ | Vandaele et al. (1998) |
| $NO_2$[a,b] | 220 K | ✗ | ✓ | Vandaele et al. (1998) |
| $O_3$[c] | 223 K | ✓ | ✓ | Serdyuchenko et al. (2014) |
| $O_3$[c,d] | 243 K | ✓ | ✓ | Serdyuchenko et al. (2014) |
| $O_4$ | 293 K | ✓ | ✓ | Thalman and Volkamer (2013) |
| Ring | | ✓ | ✓ | Chance and Kurucz (2010) |
| Polynomial | | $5^{th}$ order | $5^{th}$ order | |
| Intensity offset | | constant | constant | |

[a] $I_0$ correction is applied with SCD of $10^{17}$ molec/cm$^2$ (Aliwell et al., 2002).

[b] Orthogonalized to $NO_2$ cross-section at 298 K (Vandaele et al., 1998).

[c] $I_0$ correction is applied with SCD of $10^{20}$ molec/cm$^2$ (Aliwell et al., 2002).

[d] Orthogonalized to $O_3$ cross-section at 223 K (Serdyuchenko et al., 2014).

Several previous studies have shown that there is a systematic discrepancy between observation and model simulation of $O_4$ DSCDs (Wagner et al., 2009; Clémer et al., 2010; Wagner et al., 2011; Chan et al., 2015; Wang et al., 2016; Chan et al., 2018; Zhang et al., 2018). The discrepancies can be related to the systematic error of the $O_4$ absorption cross section, model

error, optical properties of aerosols and aerosols above the retrieval height (Ortega et al., 2016; Wagner et al., 2019). Wagner et al. (2009); Clémer et al. (2010) suggested to apply a correcting scaling factor to the measured $O_4$ DSCDs in order to bring measured and modeled results into agreement. However, the physical meaning of this scaling factor is still not fully





understood (Wagner et al., 2019). Theoretically, the optical path should be the longest under aerosol free condition for off zenith measurement. Thus, the MAX-DOAS measurement of $O_4$ DSCDs should be smaller or equal to the one simulated with pure Rayleigh atmosphere. Following the approach mentioned in Chan et al. (2019), we compared the forward simulation of $O_4$ DSCDs assuming a Rayleigh atmosphere to the MAX-DOAS observations to determine the $O_4$ scaling factor. The result

shows that the MAX-DOAS measurements occasionally exceeded the forward simulations. The monthly statistic of measured $O_4$ DSCDs exceeding the pure Rayleigh simulation is shown in Fig. A1. The ratio between simulated and measured $O_4$ DSCD can be as low as 0.70 ($2^{nd}$ percentile). The exceedances are more frequent during winter, which is mainly related to the lower aerosol optical depths in winter (see Section 4.2), while the uncertainty of surface albedo and temperature dependency of $O_4$ absorption cross section are known to be small (Wagner et al., 2019; Wang et al., 2019). In order to avoid over-correction due

to outliers, we take the $10^{th}$ percentile instead of the minimum value of the simulated and measured $O_4$ DSCD ratio as the correction factor and multiply all MAX-DOAS observations of $O_4$ DSCDs with a correction factor of 0.8. From hereafter, all $O_4$ DSCDs refer to the corrected $O_4$ DSCDs.

### 2.1.3    Aerosols and trace gases retrieval

In this study, aerosol extinction coefficient profiles are retrieved from the observations of $O_4$ DSCD at the 338 - 370 nm band us-

ing the Munich Multiple wavelength MAX-DOAS retrieval algorithm ($M^3$). As the $O_4$ DSCDs are retrieved within a relatively narrow spectral band, we can assume the wavelength dependency of the optical path within the fitting window is negligible. Thus, the forward radiative transfer simulation can be calculated at a representative wavelength of 360 nm, where the strongest $O_4$ absorption is located. A brief description of the aerosols and trace gases retrieval is presented below, a more detailed description can be found in Chan et al. (2018, 2019). The conversion of MAX-DOAS observations to aerosol extinction and trace

gases profiles requires an inversion of the underlying radiative transfer equation (Wagner et al., 2004; Hönninger et al., 2004; Sinreich et al., 2005; Frieß et al., 2006; Hartl and Wenig, 2013). The oxygen collision complex, $O_4$, has several absorption bands in the UV and VIS spectral range. Due to its known vertical distribution, the absorption signal of $O_4$, which is a combination of the concentration profile and the photon paths, which in turn are influenced by the aerosol distribution, can be used for the aerosol retrieval.

The vertical profile of the aerosol extinction coefficient is retrieved from a set of MAX-DOAS observations with different viewing directions $y(\alpha, \phi)$. A set of MAX-DOAS observations $y(\alpha, \phi)$ is defined as the $O_4$ DSCD observations at the same scanning azimuth angle $\phi$ with different elevation angles $\alpha$ within a single measurement cycle. These observations of $O_4$ DSCD are grouped together for the aerosol vertical profile retrieval. We assume that the set of measurement ($y$) can be reproduced by forward radiative transfer simulations and the forward simulations of $O_4$ DSCD are dependent on the aerosol extinction profile

($x$) and aerosol optical properties. Assuming aerosols are horizontally homogeneously distributed within the MAX-DOAS measurement range, so that the observation vector ($y$) can be described by Eq. 1.

$$y + \epsilon = f(x) + \delta \tag{1}$$





where $\epsilon$ and $\delta$ are the observation and simulation uncertainties, respectively. The aerosol extinction profile can be retrieved by fitting the forward simulations to the $O_4$ DSCD observations. In this study, all forward radiative transfer simulations were carried out using the library for Radiative transfer (libRadtran) radiative transfer model (Mayer and Kylling, 2005; Emde et al., 2016). The U.S. Standard Atmosphere (Anderson et al., 1986) mid-latitude profiles for winter (January) and summer (July) are

temporally interpolated to each month of the year for the radiative transfer calculations.

As the information contained in the observation vector $y$ is not sufficient to retrieve an unique aerosol extinction profile, the optimal estimation method is employed for the aerosol inversion. The optimal estimation approach supplemented the necessary information to the inversion in a form of an a-priori aerosol profile ($x_a$). The cost function $\chi^2$ of the retrieval can be defined by Eq. 2.

$$\chi^2 = (y - f(x))^T \cdot S_\epsilon^{-1} \cdot (y - f(x)) + (x - x_a)^T \cdot S_a^{-1} \cdot (x - x_a) \tag{2}$$

where $S_\epsilon$ represents the observation uncertainty matrix, while $S_a$ is the a-priori uncertainty covariance matrix. We assume the observations at different elevation angles are independent so that $S_\epsilon$ is a diagonal matrix. The aerosol extinction is assumed to be correlated with the neighboring layers, so that $S_a$ is defined by Eq. 3.

$$S_{a_{ij}} = \sigma_{a_i} \sigma_{a_j} exp \left( -\frac{|z_i - z_j|}{\eta_{corr}} \right) \tag{3}$$

where $z$ is the altitude of the center of the layer. Since in urban areas, aerosols are typically emitted and formed close to the surface, we assume an a-priori aerosol extinction profile following an exponentially decreasing function with a scale height of 0.5 km. The aerosol optical depth of the a-priori aerosol profile is set to 0.2, which is the average AOD measured by the co-located sun-photometer at 340 nm. The uncertainty of the a-priori aerosol profile is set to 50 % and the correlation length $\eta_{corr}$ of the aerosol inversion is assumed to be 0.5 km. As MAX-DOAS measurements are more sensitive to the aerosol and

trace gases close to the instrument, we divide the lowest 3.0 km of the troposphere unevenly into 20 layers. The lowest 1 km is divided into 10 layers with the thickness of each layer of 100 m, while the thickness of the layers between 1 km and 3 km is set to 200 m. Furthermore, we assume a fixed set of single scattering albedo of 0.95, an asymmetry parameter of 0.70 and a ground albedo of 0.04 for the radiative transfer calculations. As albedo has been reported to show only a small effect of the radiative transfer simulation of $O_4$ DSCDs (Frieß et al., 2006; Wagner et al., 2019), therefore, a fixed albedo is used for the

retrieval of all measurements. Single scattering albedo and asymmetry parameter of aerosol are the long term averages taken from the co-located sun-photometer. As the radiative transfer in the atmosphere is non-linear, therefore, the inversion of the aerosol extinction is solved iteratively by using the Gauss-Newton method.

The M[3] profile retrieval algorithm is featured with a dynamic a-priori module to avoid over-regularizing the retrieval under extreme conditions and reduce the dependency on a-priori information (Chan et al., 2019). The algorithm first use a fixed

initial a-priori (as mentioned above) to retrieve an initial aerosol profile. The fixed a-priori profile is then scaled to have the same aerosol optical depth retrieved from the initial run. The scaled a-priori is then used in the next retrieval to derive a new





aerosol extinction profile. This procedure repeats until the difference of aerosol optical depth between the new and previous result is less than 10 % or the number of iterations reaches the limit, which is set to 5 in this study.

The aerosol information obtained from the procedure described above is used for the calculation of the differential box air mass factors ΔDAMFs, required for the trace gas profile inversion. The ΔDAMFs are calculated at a single wavelength for the

retrieval of trace gas profiles using libRadtran with the Monte Carlo simulation module MYSTIC (Emde et al., 2016), assuming them to be constant within the rather narrow DOAS spectral fitting window. The relationship between ΔDAMF and DSCD can be described by the following equations.

$$\Delta DAMF_{ij} = \frac{\Delta SCD_{ij} - \Delta SCD_{zenith_j}}{\Delta VCD_j} \tag{4}$$

$$DSCD_i = \sum_j \Delta DAMF_{ij} \times \Delta z_j \times c_j \tag{5}$$

where $c_j$ is the concentration of the corresponding trace gas at the vertical layer $j$. As $NO_2$ DSCDs are retrieved at the same spectral band as $O_4$, thus, the forward simulation of ΔDAMFs for $NO_2$ profile retrieval are also calculation at the $O_4$ absorption bands of 360 nm. HCHO DSCDs are retrieved at a slightly shorter wavelength band than $O_4$. Therefore, aerosol extinction profiles obtained at 360 nm are converted to 340 nm assuming a fixed Ångström exponent (Ångström, 1929) of 1.05 for the HCHO vertical profile retrieval. This value is the annual averaged Ångström exponent calculated from the co-located

sun-photometer. The single scattering albedo (0.95), asymmetry parameter (0.70) and ground albedo (0.04) at 340 nm used for the radiative transfer calculations are assumed to be same as at 360 nm. The ΔDAMFs for the HCHO profile retrieval are then calculated using the converted aerosol profile at 340 nm. The layer settings of the trace gas profile retrieval are the same as the one used in the aerosol profile retrieval.

Following Eq. 4 and 5, a set of linear equations can be formulated by considering the measurements at different elevation

angles $\alpha_i$. Similar to the aerosol profile retrieval, the information contained in the MAX-DOAS observation is not sufficient to derive an unique solution. Therefore, the $M^3$ algorithm use the optimal estimation method (Rodgers, 2000) with a dynamic a-priori approach for the trace gas profile inversion (Chan et al., 2019). The algorithm first use a fixed initial a-priori to retrieve an initial trace gas profile. The fixed a-priori profile is then scaled to the vertical column derived in the first retrieval. The scaled a-priori is subsequently used in the next retrieval. The process iterates until the difference between retrieved and previous trace

gas column is less than 10 % or the number of iterations reaches the limit, which is set to 5 in this study.

The atmospheric layer settings of the trace gas profile retrieval are identical to the ones used in the aerosol profile retrieval. In this study, the a-priori $NO_2$ and HCHO profiles are assumed to be also exponential decreasing with a scale height of 0.5 km. The $NO_2$ vertical column density (VCD) of the a-priori is set to $1 \times 10^{16}$ molec/cm$^2$ whereas the a-priori HCHO VCD is set to $8 \times 10^{15}$ molec/cm$^2$. The vertical distribution of $NO_2$ and HCHO above the retrieval height (3 km) is assumed to follow the

U.S. Standard Atmosphere (Anderson et al., 1986).



## 2.2 Air quality monitoring network data

Ambient $NO_2$ and $PM_{10}$ (particulate matter with diameter smaller than $10\,\mu$m, typically given as mass concentration in $\mu$g/m$^3$) data in Munich are acquired from an ambient air quality monitoring station operated by the Bavarian State Ministry of the Environment and Consumer Protection. The station is 1.2 km north west to the MAX-DOAS measurement site (48.155°N,

11.555°E) (blue dot in Fig. 1). Ambient $NO_2$ mixing ratios are measured by an in-situ chemiluminescence $NO_2$ analyzer, while $PM_{10}$ concentrations are measured with a beta attenuation and light scattering based in-situ particle analyzer. Details of the air quality monitoring network as well as air quality monitoring data can be found on the website of the European Environment Agency (https://www.eea.europa.eu/).

## 2.3 Sun-photometer measurements

A sun-photometer (CIMEL Electronique, CE-318) is installed next to the 2D MAX-DOAS instrument, providing multi-wavelength measurements of aerosol optical properties (Holben et al., 2001). As part of the AERosol RObotic NETwork (AERONET) (Holben et al., 1998), instrument #198, data include measurements at 7 different wavelengths, which are 340, 380, 440, 500, 675, 870 and 1020 nm, and aerosol optical properties are retrieved by an automated inversion algorithm developed by Dubovik and King (2000); Dubovik et al. (2006). Cloud screened and quality assured Level 2.0 data are used in this

study.

## 2.4 OMI satellite observations

The Ozone Monitoring Instrument (OMI) is a passive nadir-viewing satellite borne push-broom imaging spectrometer (Levelt et al., 2006) on board of the Earth Observing System's (EOS) Aura satellite. The Aura satellite was launched on 15 July 2004, orbiting at an altitude of ∼710 km with a local equator crossing time of 13:45 on ascending node. The OMI instrument consists

of two CCD arrays covering a wavelength range from 264 nm to 504 nm. Each scan provides measurements of earthshine radiance at 60 positions across the orbital track covering a swath of approximately 2600 km. The spatial resolution of OMI varies from 13 km (across-track) × 24 km (along-track) at nadir to 160 km (across-track) × 40 km (along-track) at the edges of the swath. OMI scans along 14.5 sun-synchronous polar orbits per day providing daily global coverage.

The OMI $NO_2$ products derived within the framework of the quality assurance for the essential climate variables (QA4ECV)

project are used in this study (Boersma et al., 2018). $NO_2$ SCDs are derived from earthshine radiance spectra in the visible band from 405 - 465 nm using a DOAS retrieval. The SCDs are then converted to vertical column densities (VCDs) using the concept of air mass factors (AMFs) (Solomon et al., 1987). The AMFs used in the QA4ECV OMI $NO_2$ product are calculated at 437.5 nm with $NO_2$ vertical profiles taken from the global chemistry transport model TM5-MP (Williams et al., 2017). Albedo data is from the climatology albedo database derived from 5 years of OMI observations (Kleipool et al., 2008). Separation of

stratospheric and tropospheric columns, which is necessary to provide proper information for the AMF calculation, is achieved by the model assimilation approach (Dirksen et al., 2011).





## 2.5 TROPOMI satellite observations

The TROPOspheric Monitoring Instrument (TROPOMI) is a passive nadir viewing satellite borne push-broom imaging spectrometer on board the Copernicus Sentinel 5 Precursor (S5P) satellite. The satellite was launched on 13 October 2017 on a sun-synchronous orbit at an altitude of $\sim$824 km with a local equator overpass time of 13:30 on ascending node. The instru-
ment has 8 spectral bands covering UV, VIS, near infrared (NIR) and short-wavelength infrared (SWIR). The instrument takes measurements at 450 positions across the orbital track which cover a swath of $\sim$2600 km, providing daily global coverage observations. The spatial resolution of the instrument is 3.6 km (across-track) $\times$ 7.2 km (along-track) for measurements taken before 6 August 2019. Thereafter the instrument was switched to a better spatial resolution of 3.6 km (across-track) $\times$ 5.6 km (along-track). A more detailed description of the TROPOMI instrument can be found in Veefkind et al. (2012).
The operational TROPOMI $NO_2$ and HCHO products are used in this study (van Geffen et al., 2019; De Smedt et al., 2018). The operational TROPOMI $NO_2$ retrieval algorithm is very similar to the OMI product as demonstrated in the QA4ECV project. The operational TROPOMI HCHO product retrieves HCHO SCDs with a large fitting window of 328.5 - 359 nm. The retrieved SCDs are then converted to VCDs using the AMF approach. The AMFs are calculated at 340 nm using HCHO vertical profiles from the global chemistry transport model TM5-MP. Similar to the operational $NO_2$ product, albedo data are taken
from the OMI climatology and will be updated to TROPOMI albedo product when it is available. A more detailed description of the TROPOMI HCHO retrieval algorithm can be found in (De Smedt et al., 2018).
    We have regridded and calculated the annual average of TROPOMI tropospheric $NO_2$ and HCHO VCDs over Germany and its surrounding regions. The annual averaged TROPOMI tropospheric $NO_2$ and HCHO maps are shown in Fig. 3. The location of Munich is indicated by the circle marker in Fig. 3. Significant $NO_2$ hot spots can be observed over major cities indicating the significant contribution of anthropogenic emissions. On the other hand, the spatial distribution of HCHO is rather
homogeneous due to strong natural emissions.

## 3 $NO_2$ and HCHO retrievals for Munich

### 3.1 Spatial variability

In order to investigate the spatial variability of $NO_2$ and HCHO, we utilize the azimuthal scans of the MAX-DOAS. Tropo-
spheric $NO_2$ VCD measured by the MAX-DOAS with different viewing azimuth angles are plotted in Fig. 4. Fig. 4a shows the measurements in winter (December, January and February), while measurements in summer (June, July and August) are shown in Fig. 4b. $NO_2$ columns measured with different viewing azimuth angles during winter show a rather homogeneous spatial distribution, while during summer $NO_2$ columns show slightly higher values ($\sim$20 %) in the south and lower values in the north. A more homogeneous distribution of $NO_2$ is due to better mixing of $NO_2$ within the mixing layer during win-
ter. The average wind speed in Munich during winter is $\sim$12 km/h, while the summertime average wind speed is $\sim$8 km/h (https://www.en.meteo.physik.uni-muenchen.de/wetter/index.html). Stronger wind speed together with the shallower mixing layer as known from ceilometer measurements results in better mixing of $NO_2$ during winter. In addition, the atmospheric

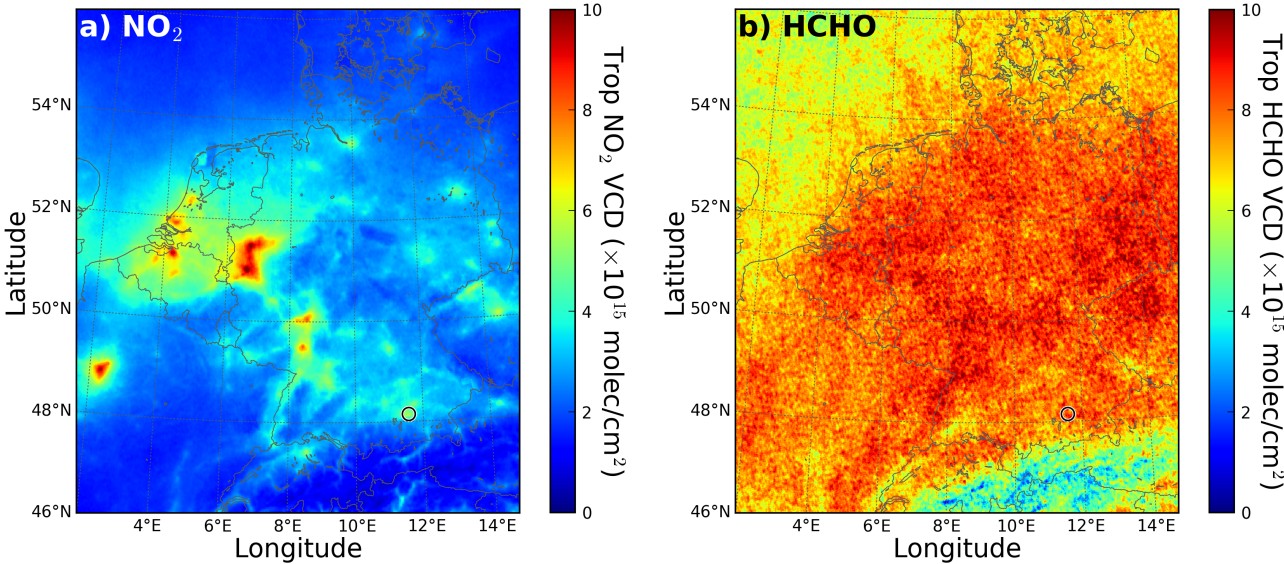

**Figure 3.** Annual average TROPOMI tropospheric (a) $NO_2$ and (b) HCHO VCDs over Germany and its surrounding regions. The circle marker indicates the location of Munich. Data from May 2018 to April 2019 are used in the calculation of average maps. Values above or below the range of the color scale are set to the maximum or minimum value of the color scale.

lifetime of $NO_2$ is longer in winter due to lower photolysis rate which also leads to more homogeneous $NO_2$ distribution. In the south of the measurement site where the city center is located, higher $NO_2$ levels are observed during summer. There are several local emission hot spots in the city, such as a number of busy crossroads and a minor natural gas power plant. In addition, lower wind speed and shorter lifetime of $NO_2$ reduced the dispersion in summer, thus resulting in rather inhomogeneous

$NO_2$ distribution.

Fig. 4c and d show the MAX-DOAS measurements of HCHO VCD for different azimuth angles. Measurements taken during winter and summer are shown in Fig. 4c and d, respectively. In contrast to the $NO_2$ distribution, the spatial distribution of HCHO is more homogeneous in summer, while higher values are observed in the south and south west of the measurement site during winter. Homogeneous distribution of HCHO during summer is likely related to its source characteristic. A large

fraction of HCHO and its precursors is related to biogenic emissions from vegetation in summer. These biogenic sources are areal sources and widely distributed over the city and its surrounding areas. One of the major biogenic emission sources is the English Garden, which is a public park with an area of $3.7 \, km^2$ and located in the center of Munich. Therefore, the spatial distribution of HCHO is expected to be more homogeneous in summer. Biogenic emissions are greatly reduced in winter and anthropogenic point sources, i.e., the natural gas power plant and domestic wood-burning heating system installed in old

buildings, become the dominant source. Therefore, slightly elevated HCHO values are observed in the south and south west

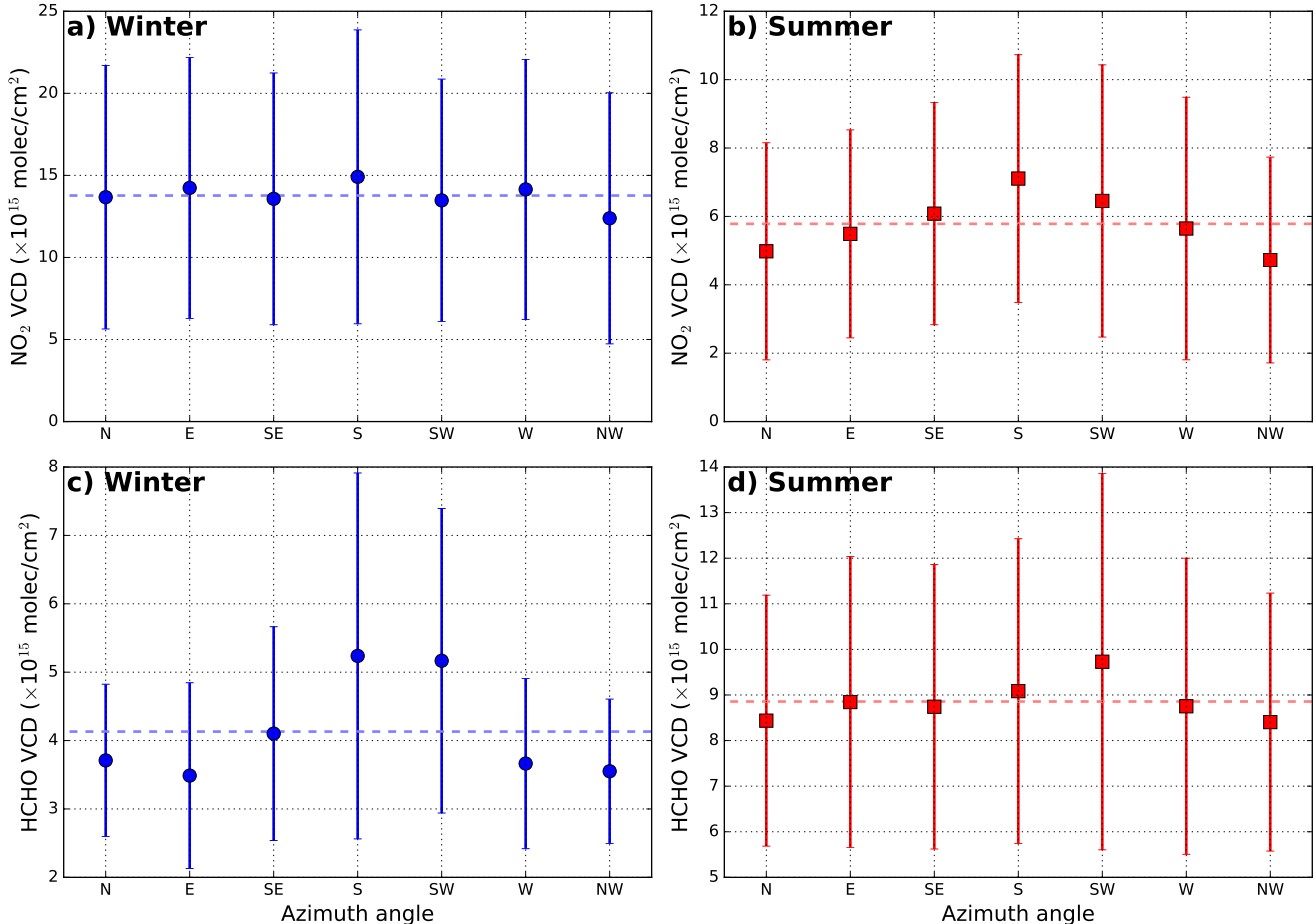

**Figure 4.** MAX-DOAS measurements of $NO_2$ VCD for different viewing azimuth angles for (a) winter (December, January and February) and (b) summer (June, July and August). HCHO VCDs measured in winter and summer are shown in (c) and (d), respectively. Error bars indicate the $1\sigma$ standard deviation variation range. Note that the scale of y-axis of each subplot is different.

direction during both summer winter. Less pronounced HCHO peak in the south during summer is likely related to the reduced domestic heating.

## 3.2 Day of week variability

Human activities usually fall into a 7-days weekly cycle. Reduction of industrial activities as well as traffic volume during weekend lead to lower levels of pollutant emission, an effect known as the weekend effect (Cleveland et al., 1974). We have investigated the weekend effect of $NO_2$ and HCHO using the MAX-DOAS measurements in Munich. Fig. 5 shows the normalized mean weekly cycle of $NO_2$ and HCHO. Data are normalized by dividing by the weekday mean value (Monday to

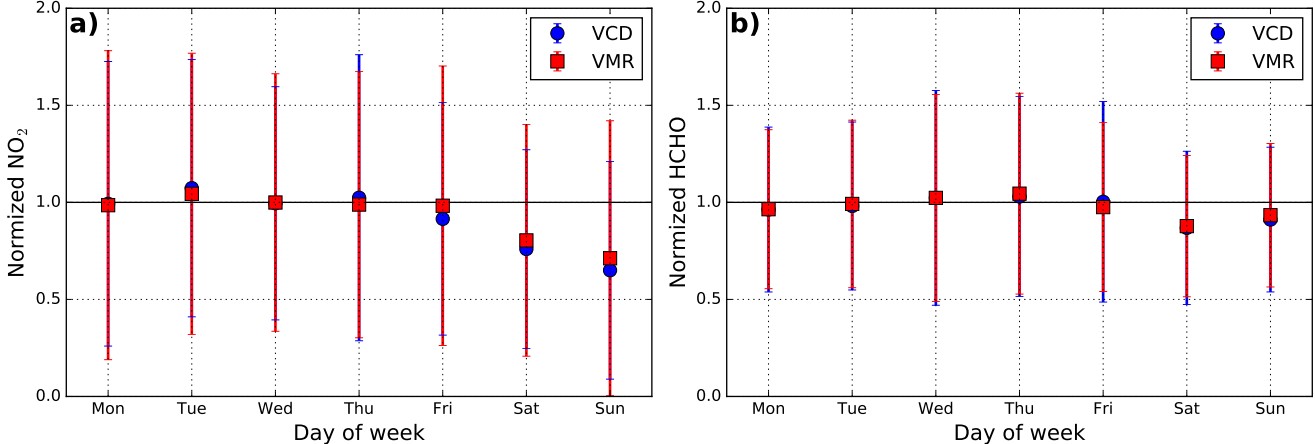

**Figure 5.** Average weekly cycle of (a) NO$_2$ and (b) HCHO measured by the MAX-DOAS instrument. Data are normalized by dividing by the mean weekday value (Monday to Friday). Both vertical column (blue circle) and surface mixing ratio (red square) are shown. Error bars indicate the 1 $\sigma$ standard deviation variation range.

Friday). Both vertical column and surface mixing ratio are shown. In the case of NO$_2$ vertical columns and surface mixing ratios show lower values during weekends. NO$_2$ VCDs are reduced by 25 % and 35 % on Saturday and Sunday, respectively. The weekend reduction of surface mixing ratios is similar to that of the vertical column with reductions of ∼20 % for Saturday and 30 % for Sunday. The reduction of NO$_2$ level during weekend implies a large anthropogenic contribution of NO$_2$ emissions.

Compared to NO$_2$, the weekend reduction effect of HCHO is less pronounced. HCHO vertical columns and surface mixing ratios are reduced by 13 % and 9 % for Saturday and Sunday, respectively. As natural emission, such as biogenic emission from plants, do not show a weekly pattern, the reduction during the weekend suggests that anthropogenic emissions of HCHO and its precursors have a substantial (>10 %) contribution to the ambient VOCs.

### 3.3   Relations among aerosol, NO$_2$ and HCHO

The correlations among aerosol extinction coefficients, NO$_2$ and HCHO mixing ratios can be used to investigate the composition and sources of aerosols (Veefkind et al., 2011). Fossil fuel combustion is the most significant primary source of NO$_2$ and aerosols, while HCHO correlates strongly with secondary organic aerosol formation. Fig. 6a shows the correlation between surface aerosol extinction coefficients ($\varepsilon_{\mathrm{surf}}$) and NO$_2$ mixing ratios, while the correlation between $\varepsilon_{\mathrm{surf}}$ and HCHO mixing ratios is shown in Fig. 6b. Aerosol extinction coefficients, NO$_2$ and HCHO mixing ratios at the lowest layer of the MAX-DOAS

profile are used in the analysis. Considering the meteorological influences are very different during different seasons, we have separated measurements during summer and winter. Both surface NO$_2$ and HCHO mixing ratios show significant correlation with $\varepsilon_{\mathrm{surf}}$ with correlation coefficients ranging from $0.39 \leq R \leq 0.73$ with better correlations observed during winter. The aerosol extinction to NO$_2$ ratio for summer and winter is very similar. Assuming primary aerosols and NO$_2$ originate from





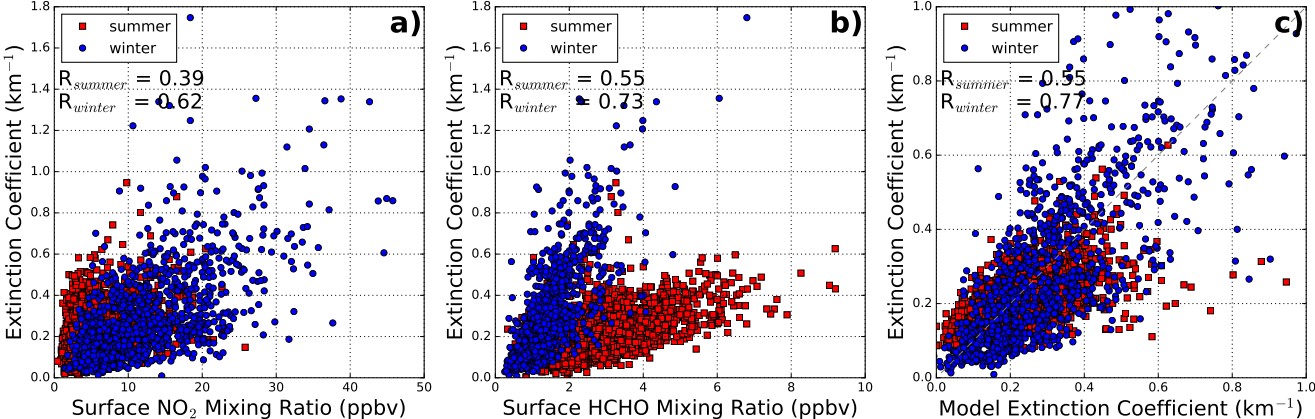

**Figure 6.** (a) shows the correlation between surface aerosol extinction coefficients ($\varepsilon_{\mathrm{surf}}$) and $NO_2$ mixing ratios. (b) shows the correlation between $\varepsilon_{\mathrm{surf}}$ and HCHO mixing ratios. (c) shows the scatter plot of measured and modeled surface aerosol extinction coefficients. Data measured in summer (June, July and August, red markers) and winter (December, January and February, blue markers) are shown.

the same sources, a rather constant aerosol to $NO_2$ ratio indicates the sources of primary emission are similar in summer and winter. On the other hand, a higher aerosol extinction to HCHO ratio is observed during winter compared to summer ratio. Higher aerosol extinction to HCHO ratio reflects a longer atmospheric lifetime of secondary aerosol and HCHO in winter, whereas higher photolysis rates in summer result in a lower aerosol to HCHO ratio.

Assuming that $NO_2$ mixing ratios are related to primary emissions of aerosols and HCHO mixing ratios are related to secondary aerosol formation, we used a multiple linear regression model to estimate the contribution of primary and secondary aerosols. The comparison of modeled and MAX-DOAS measurements of aerosol extinction coefficients is shown in Fig. 6c. The Pearson correlation coefficients ($R$) between modeled and measured aerosol extinction coefficient for summer and winter are 0.55 and 0.77, respectively. The model only considered primary and secondary sources of aerosol, while factors, such as

pollution transport and meteorological effects are not considered. Better correlation in winter indicates larger contributions of primary and secondary aerosols, whereas transportation and meteorological effects show a stronger influence on the ambient aerosol concentrations. Better correlation between aerosol and HCHO implies the large contribution of secondary aerosol, while primary aerosol sources show a less significant contribution as indicated by the correlation between aerosol and $NO_2$.

## 4 Intercomparison of MAX-DOAS retrievals with other data-sets

### 4.1 Comparison of surface aerosol and $NO_2$ concentrations

Aerosol extinction coefficients are related to the particle concentrations in the atmosphere, depending on the aerosol composition and size distribution. Thus, as a first approximation assuming constant composition, we compare the aerosol extinction



coefficients at the lowest layer ($\varepsilon_{\mathrm{surf}}$) of the MAX-DOAS profile retrieval to $PM_{10}$ concentrations reported from the nearby air quality monitoring station. Time series of $\varepsilon_{\mathrm{surf}}$ at 360 nm and $PM_{10}$ concentrations are shown in Fig. 7a. As the in-situ air quality monitoring station only provides hourly data, MAX-DOAS measurements for all azimuth directions are averaged to hourly and monthly data for comparison. Both, $\varepsilon_{\mathrm{surf}}$ and $PM_{10}$, show similar variation pattern with slightly higher values in

winter, however, $\varepsilon_{\mathrm{surf}}$ varies in a wide range with hourly value ranging from $0.005\,\mathrm{km^{-1}}$ up to $1.859\,\mathrm{km^{-1}}$. The correlation between MAX-DOAS measurements of $\varepsilon_{\mathrm{surf}}$ and $PM_{10}$ concentrations from in-situ measurement is shown in Fig. 7b. Monthly averaged MAX-DOAS data show a reasonable agreement with the in-situ $PM_{10}$ measurements with $R$ of 0.66. This moderate correlation can be explained by the differences in physical quantities of the two measurements. The aerosol extinction coefficient is not only related to the aerosol mass concentration, but also strongly related to the micro-physical properties of aerosol,

such as the size distribution and particle composition. Meteorological factors, such as, humidity and temperature, could have big impacts on the aerosol size distribution and optical properties. Therefore, the relation between $\varepsilon_{\mathrm{surf}}$ and $PM_{10}$ concentrations can be very different in different seasons (Schäfer et al., 2008). In addition, the spatial coverage of the two measurements is quite different. MAX-DOAS observations typically cover a few kilometers around the measurement site, whereas the in-situ measurements are only representative for the small area surrounding the station and governed by local conditions (see e.g. Geiß

et al., 2017).

    We have also compared the surface $NO_2$ mixing ratios retrieved from the MAX-DOAS observations to the in-situ monitor, the corresponding time series are shown in Fig. 7c. The MAX-DOAS surface $NO_2$ mixing ratios are taken from the lowest layer of the $NO_2$ vertical profile retrieval. Similar to the $PM_{10}$ comparison, individual MAX-DOAS surface $NO_2$ data are averaged to hourly and monthly values for comparison. The surface $NO_2$ mixing ratios show a similar seasonal pattern as $PM_{10}$ with higher

values during winter and lower in summer. The surface $NO_2$ mixing ratios vary in a wide range. Hourly averaged MAX-DOAS data is ranging from 0.4 ppbv up to 53.5 ppbv, while in-situ monitor reports a variation of 1.3 - 100.2 ppbv. The MAX-DOAS observations are systematically lower than the in-situ monitor by $\sim$50 %. Fig. 7d shows the scatter plot between MAX-DOAS and in-situ measurements of surface $NO_2$ mixing ratios. Both hourly and monthly averaged data show good agreement with $R = 0.91$ for the monthly values. The slope of the total least squares regression line is 0.54 with an offset of 0.61 ppbv. Lower

values measured by the MAX-DOAS are mainly due to the differences in vertical coverage. $NO_2$ mixing ratios at the lowest layer of the MAX-DOAS retrieval represent the average values from 20 m (roof top level) to 120 m above ground, while the in-situ monitor measures at $\sim$15 m above street level. The major source of $NO_2$ in urban areas are traffic emissions which are emitted at street level, therefore, the atmospheric concentration of $NO_2$ is expected to be lower after being dispersed to upper altitudes. In addition, the MAX-DOAS reports $NO_2$ mixing ratios averaged along a long optical path, which covers residential

areas and city parks, where the $NO_2$ mixing ratios are expected to be lower. As a consequence, the MAX-DOAS is in general measuring lower surface $NO_2$ mixing ratios than the in-situ monitoring station.

## 4.2   Comparison of aerosol optical depth

Time series of AOD at 360 nm over Munich derived from MAX-DOAS and sun-photometer measurements are compared in Fig. 8a. For this purpose sun-photometer data measured at 340 and 380 nm have been interpolated to 360 nm following the





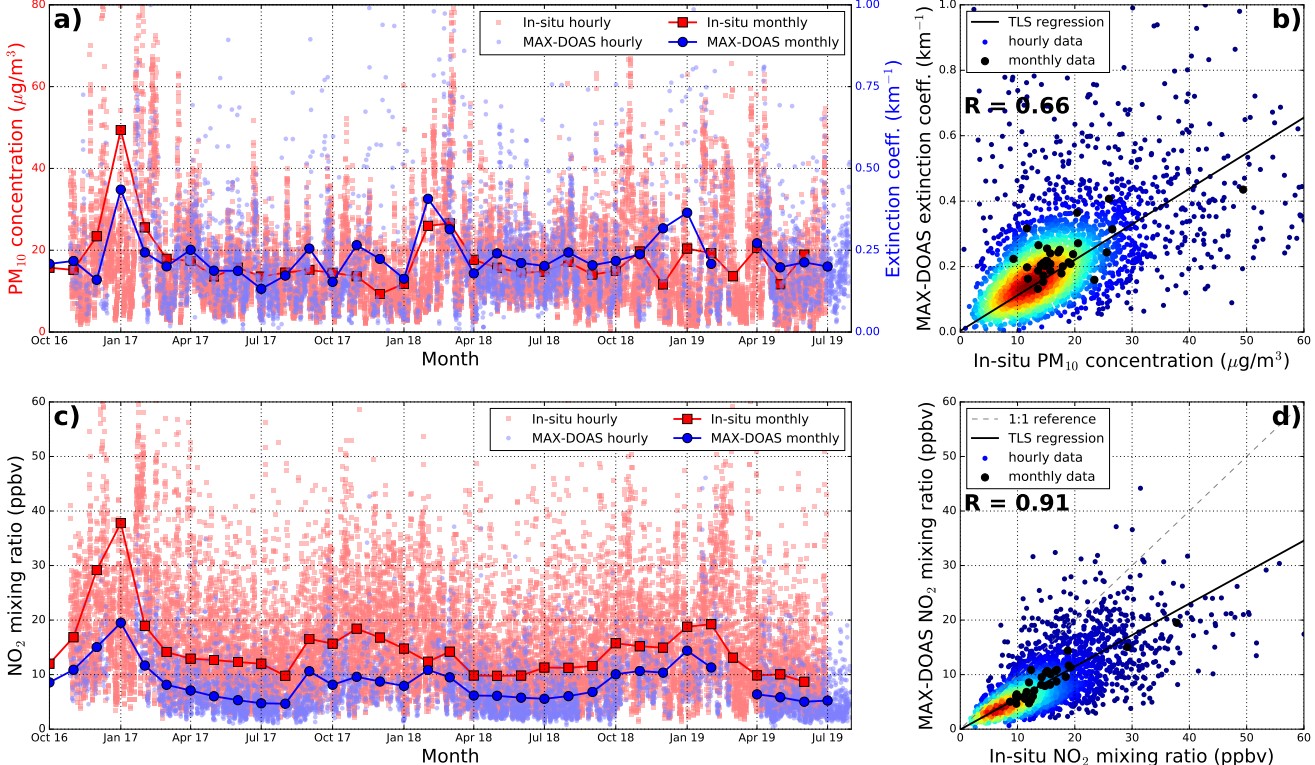

**Figure 7.** (a) Time series of surface aerosol optical extinction coefficients $\varepsilon_{surf}$ at 360 nm retrieved from MAX-DOAS observations (blue curve) and $PM_{10}$ concentrations measured by the air quality monitoring station (red curve). (b) Scatter plot of $\varepsilon_{surf}$ against $PM_{10}$ concentrations. (c) Time series of surface $NO_2$ mixing ratios measured by the MAX-DOAS (blue curve) and the air quality monitoring station (red curve). (d) Scatter plot of the $NO_2$ mixing rations measured by the MAX-DOAS against the air quality monitoring station measurements. Correlation coefficient and total least squares regression lines are calculated based on monthly averaged data.

Ångström exponent approach. As the temporal resolution of the MAX-DOAS and the sun-photometer are different, individual data are averaged to hourly and monthly values for comparison. Missing data are due to cloud filtering or instrument maintenance. The annual average of the AOD from the MAX-DOAS and sun-photometer observations is 0.21 and 0.23, respectively, indicating consistency between both remote sensing techniques. In contrast to the surface aerosol extinction coefficients and

5    $PM_{10}$ concentrations the annual cycle of the AOD shows larger values in summer and lower values in winter. This is true for measurements from MAX-DOAS and sun-photometer. Stronger convection and insolation resulting in extended mixing layers as well as enhanced emission of e.g. biogenic VOCs are the main reasons of increased AOD in spring and summer. Moreover, long range transport of Saharan dust occurs frequently with sometimes exceptionally large contributions (Ansmann et al., 2003; Wiegner et al., 2011). The difference between the annual variation of $\varepsilon_{surf}$ and AOD suggests a different vertical distribution





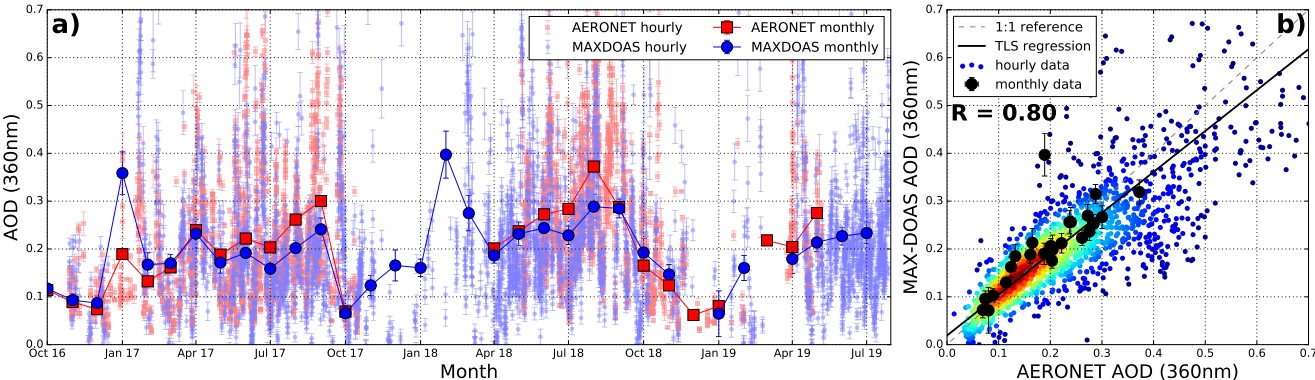

**Figure 8.** (a) Time series of aerosol optical depth measured by MAX-DOAS (blue curve) and sun-photometer (red curve). (b) scatter plot of aerosol optical depth measured by MAX-DOAS and sunphotometer. The correlation coefficient and total least squares regression line is calculated based on the hourly averaged data.

of aerosols in different seasons. Due to reduced vertical exchange aerosols are concentrated near the surface during winter, resulting in increased $\varepsilon_{\mathrm{surf}}$ and low AOD.

The scatter plot of MAX-DOAS and sun-photometer derived AOD is shown in Fig. 8b. Hourly averaged data correlate well with $R = 0.80$. Despite the high correlation between the two data-sets, AOD derived from MAX-DOAS is in general slightly

lower than the AERONET retrievals, especially under high aerosol load. The slope of the total least squares regression line is 0.86 with an offset of 0.02. The discrepancy between the results can be explained by the differences in the measurement techniques: the MAX-DOAS retrieval derives the AOD from observations of $O_4$ absorption, and is mostly sensitive to aerosols in the lowest few kilometers of the troposphere as it uses lower elevation angles. In contrast the sun-photometer retrieval is based on the reduction of the transmission of solar radiation along the line of sight, thus covering the full vertical extent

of the atmosphere. As the AODs reported from the MAX-DOAS only represent the AODs of the lowest 3 km, while the sun-photometer AODs cover the entire atmosphere, lower AODs observed by the MAX-DOAS is expected. Furthermore, the assumptions on aerosol optical properties in the MAX-DOAS retrieval also contribute to the uncertainties of the MAX-DOAS AOD ($\sim5\,\%$) (Chan et al., 2019).

## 5   MAX-DOAS retrievals for satellite validation

### 5.1   Comparison of NO$_2$ columns

Tropospheric NO$_2$ vertical column densities retrieved from the MAX-DOAS measurements are compared to OMI and TROPOMI observations over Munich. MAX-DOAS VCDs are temporally averaged around the OMI and TROPOMI overpass time of 12:00 - 14:00 (local time), while OMI and TROPOMI data are spatially averaged for pixels within 10 km of the MAX-DOAS





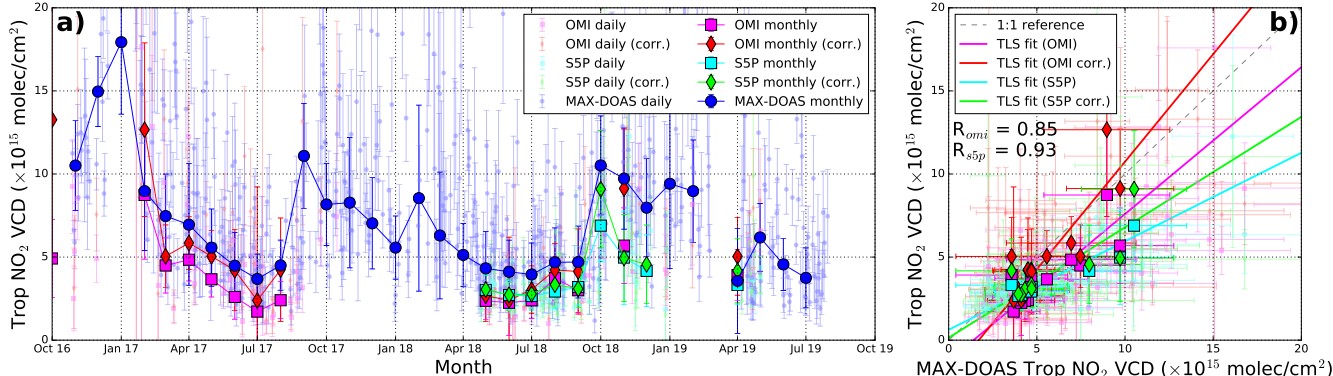

**Figure 9.** (a) Time series of tropospheric $NO_2$ vertical column densities measured by MAX-DOAS, OMI and TROPOMI (labeled as S5P). MAX-DOAS data are temporally averaged around the satellite overpass time, while OMI and TROPOMI observations are spatially averaged within 10 km of the MAX-DOAS measurement site. OMI and TROPOMI VCDs retrieved using MAX-DOAS profile as a-priori information are shown as well (labeled as OMI corr. and S5P corr.). (b) scatter plot of tropospheric $NO_2$ VCDs measured by MAX-DOAS and TROPOMI.

measurement site. Time series of tropospheric $NO_2$ VCDs from MAX-DOAS, OMI and TROPOMI observations are shown in Fig. 9a. OMI and TROPOMI $NO_2$ VCDs retrieved using MAX-DOAS profile as a-priori information are also indicated. Daily and monthly averages are shown. Missing data are due to high cloudiness, pixel anomaly of OMI or maintenance of the MAX-DOAS instrument. Both ground based and satellite measurements show lower $NO_2$ VCDs in summer and higher values during winter. Higher $NO_2$ levels in winter are mainly due to higher emissions, e.g., domestic heating, and longer atmospheric

life time of $NO_2$.

The scatter plot of OMI and TROPOMI observations of tropospheric $NO_2$ VCDs against MAX-DOAS measurements is shown in Fig. 9b. Both OMI and TROPOMI $NO_2$ observations show good correlation with MAX-DOAS measurements with $R = 0.85$ and $R = 0.93$, respectively. However, both space borne observations report lower $NO_2$ columns than the MAX-DOAS.

Averaged difference between OMI and TROPOMI satellite observations and MAX-DOAS measurements of $NO_2$ VCDs are $-2.32 \times 10^{15}$ molec/cm$^2$ and $-2.25 \times 10^{15}$ molec/cm$^2$, respectively. The underestimation of $NO_2$ VCDs is partly related to the a-priori vertical distribution profile of $NO_2$ used in the air mass factor calculation of the satellite retrieval. These satellite a-priori profiles are taken from the TM5 chemistry transport model simulation. The horizontal resolution of TM5 is rather coarse ($1° \times 1°$) which is not able to fully resolve emission hot spots over cites. In order to quantify the influence of the a-priori $NO_2$

profile in the satellite retrieval, we have recomputed the OMI and TROPOMI $NO_2$ VCDs by using MAX-DOAS $NO_2$ profiles as a-priori information (labeled as OMI corr. and S5P corr. in Fig. 9). Monthly averages of the a-priori $NO_2$ profiles used in the satellite retrieval are shown in Fig. 10a, while the corresponding MAX-DOAS retrievals are shown in Fig. 10b. MAX-DOAS $NO_2$ profiles show about 4 times higher $NO_2$ levels at the surface compared to the original a-priori profiles used in the satellite retrieval. Using the MAX-DOAS $NO_2$ profiles as a-priori information generally increased the OMI and TROPOMI $NO_2$ VCDs

by ∼45 % and ∼17 %, respectively. Due to difference in temporal coverage, OMI provides longer term measurement while





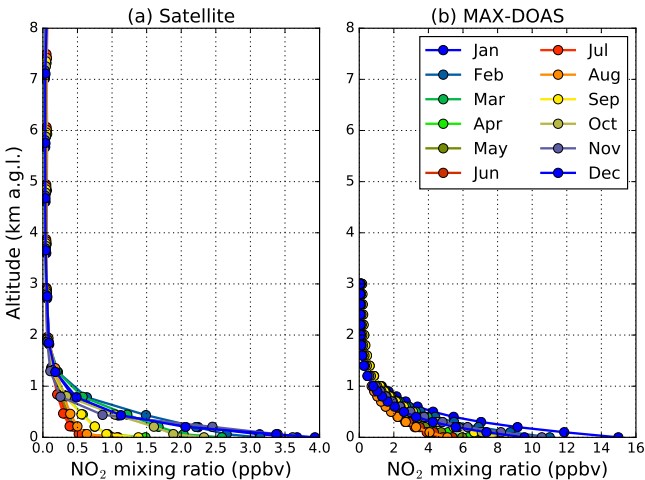

**Figure 10.** (a) Monthly average of $NO_2$ a-priori profiles used in the satellite retrieval. (b) MAX-DOAS measurements of $NO_2$ profiles.

TROPOMI measurements are only available after November 2017, the percentage increase of OMI and TROPOMI $NO_2$ VCDs are quite different. If we only consider the same period of November 2017 to July 2019, the percentage of increase is similar for OMI and TROPOMI. As can be seen in Fig. 9 the absolute values of OMI and TROPOMI $NO_2$ VCDs retrieved with MAX-DOAS $NO_2$ profiles as a-priori agree better with the MAX-DOAS measurements with correlation nearly unchanged.

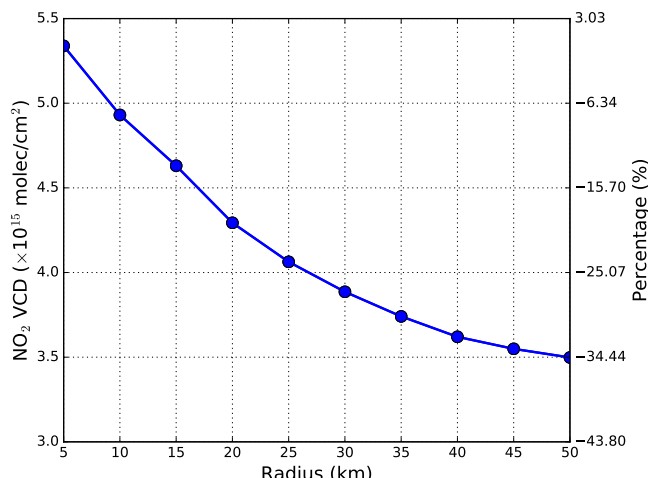

**Figure 11.** Tropospheric $NO_2$ VCDs measured by TROPOMI spatially averaged with different radius surrounding the MAX-DOAS measurement site.





Previous satellite observations often underestimated the tropospheric $NO_2$ columns over cities or pollution hot spots. The underestimation is partially related to the large satellite footprint which is not able to capture the spatial gradient of $NO_2$ due to the averaging over large satellite pixels (Wenig et al., 2008; Chan et al., 2012). This averaging effect over hot spots can be estimated by using high resolution TROPOMI observations. Tropospheric $NO_2$ VCDs measured by TROPOMI are spatially

averaged with different radii are shown in Fig. 11. Satellite data with their pixels center coordinate within certain radius of the MAX-DOAS measurement site are used in the spatial averaging. The MAX-DOAS measurement in the UV typically covers a range of 5 - 8 km depending on the visibility, while the measurement in the VIS has a better coverage of 8 - 12 km. Therefore, the percentage of underestimation relative to the 5 km average is shown in Fig. 11 as reference. The result shows that the averaged $NO_2$ VCDs decreases with increasing averaging radius. $NO_2$ columns are underestimated by ∼8 % with an averaging radius

of 10 km which is approximately the size of OMI footprint at nadir (13 km × 24 km). The underestimation increases to ∼13 % and ∼34 % for averaging radius of 15 km (average OMI pixel size) and 50 km. These numbers are characteristic for pollution hot spots of the size of Munich (approximately 5 km in radius), but they could be different for hot spots of different size and spatial distribution. Although the spatial resolution of TROPOMI observations have been significantly improved compared to its predecessors, satellite observations are still critical to resolve spatial features of pollutant within a city. Therefore, ground

based measurements are essential for the investigation of small scale pollution within a city.

## 5.2  Comparison of HCHO columns

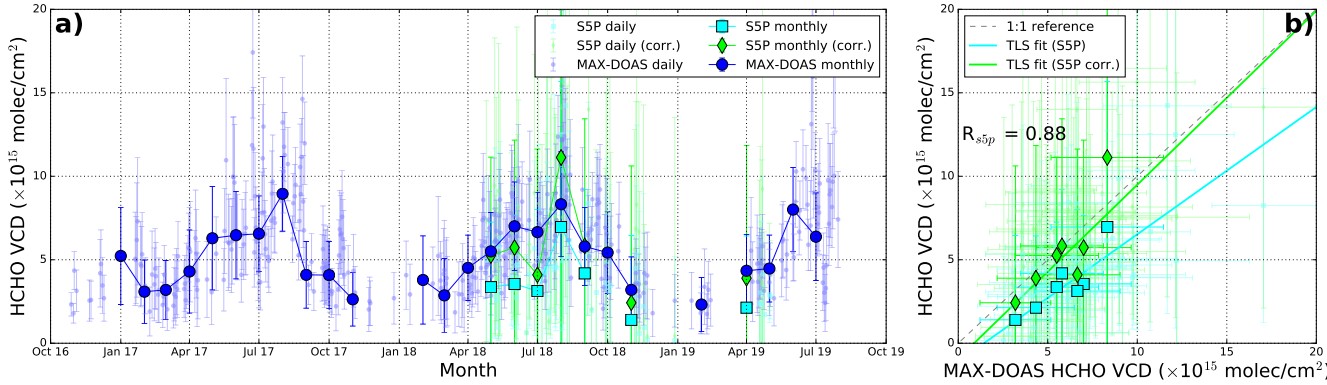

**Figure 12.** (a) Time series of tropospheric HCHO vertical column densities measured by MAX-DOAS and TROPOMI. MAX-DOAS data are temporally averaged around the TROPOMI overpass time, while TROPOMI observations are spatially averaged within 10 km of the MAX-DOAS measurement site. TROPOMI VCDs retrieved using MAX-DOAS profile as a-priori information are also indicated (labelled "corr"). (b) scatter plot of tropospheric HCHO VCDs measured by MAX-DOAS and TROPOMI.

MAX-DOAS observations of HCHO VCDs are also used to validate TROPOMI measurements. Time series of HCHO VCDs measured by the MAX-DOAS and TROPOMI are shown in Fig. 12a. MAX-DOAS VCDs are temporally averaged around the



TROPOMI overpass time of 12:00 - 14:00 (local time), while TROPOMI data are spatially averaged for pixels within 10 km of the MAX-DOAS measurement site. Again, gaps are mainly due to high cloud amount and maintenance of the MAX-DOAS instrument. In contrast to the $NO_2$ data, the HCHO VCDs show higher values in summer and lower VCDs during winter. Higher HCHO levels are expected in summer due to stronger biogenic emissions of precursor VOCs from vegetation and

higher oxidation rate of VOCs. The scatter plot of TROPOMI HCHO VCDs against MAX-DOAS measurements is shown in Fig. 12b. Satellite and ground based measurements show good correlation with $R = 0.88$ for monthly averaged HCHO VCDs. The absolute values measured by TROPOMI is however ∼30 % lower than the MAX-DOAS measurements. The average HCHO VCDs measured by TROPOMI and MAX-DOAS are $4.42 \times 10^{15}$ moles/cm$^2$ and $6.56 \times 10^{15}$ moles/cm$^2$, respectively. The slope of the total least squares regression line is 0.76 with an offset of $-1.10 \times 10^{15}$ moles/cm$^2$. Analogously to the previous

section we have recomputed the TROPOMI HCHO VCDs by using MAX-DOAS profiles (see Fig. 13b) instead of the TM5-profiles (see Fig. 13a) as a-priori information to estimate the influence of the a-priori profile. The MAX-DOAS profiles show larger amounts of HCHO in the lower troposphere. Using the MAX-DOAS profile as a-priori in the satellite retrieval in general enhanced the HCHO columns by ∼50 %. The averaged TROPOMI HCHO VCD increased to $6.37 \times 10^{15}$ moles/cm$^2$. The slope of the regression line of the new data-set also increased to 1.04 (see Fig. 12b).

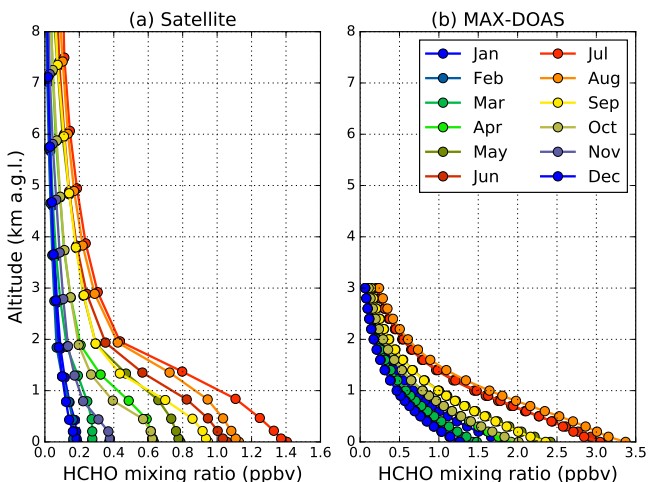

**Figure 13.** (a) Monthly averages of HCHO mixing ratio based on TM5-simulations as used a-priori profiles in the satellite retrieval. (b) MAX-DOAS measurements of HCHO profiles.

Similar to the discussion on $NO_2$ retrievals we have analyzed the spatial averaging effect of satellite observations over a HCHO emission hot spot. TROPOMI HCHO VCDs are spatially averaged with different radii surrounding the MAX-DOAS measurement site and the result is shown in Fig. 14. The underestimation relative to the 5 km average is also shown on the right axis. As expected the averaged HCHO VCDs decrease with increasing averaging radius. HCHO column for an averaging radius of 10 km is ∼7 % lower than the 5 km average. The underestimation increases to ∼8 % and ∼15 % with an averaging radius





of 15 km and 50 km, respectively. The decrease pattern indicated that there are significant anthropogenic HCHO or HCHO precursor emission in Munich. However, compared to the decreasing pattern of $NO_2$, HCHO shows a more homogeneous distribution as it is mainly originated from regional sources.

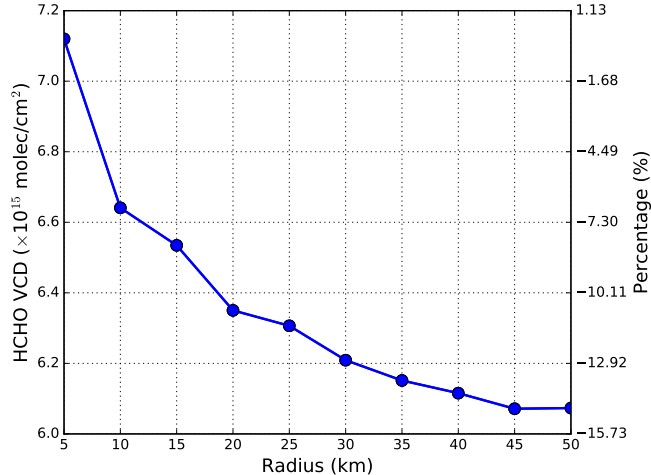

**Figure 14.** Tropospheric HCHO VCDs measured by TROPOMI spatially averaged with different radii surrounding the MAX-DOAS measurement site.

## 6 Summary and conclusion

In this paper, we present the first 2D Multi-AXis Differential Optical Absorption Spectroscopy (MAX-DOAS) observations of nitrogen dioxide ($NO_2$) and formaldehyde (HCHO) vertical profile in Munich, Germany. The measurement covers the time period from October 2016 to July 2019. We have determined vertical columns and vertical profiles of the aerosol extinction coefficient, $NO_2$ and HCHO for Munich. The measured data are used to analyze the spatio-temporal variation of $NO_2$ and HCHO. The spatial distribution of $NO_2$ was in general quite homogeneous in winter, however, with higher values at the city center during summer. Spatial pattern of HCHO shows higher values in the south in winter and a rather homogeneous distribution in summer. Analysis of the relations between aerosols, $NO_2$ and HCHO shows higher aerosol to HCHO ratios in winter indicating a longer atmospheric lifetime of aerosol and HCHO and suggests that secondary aerosol formation is the major source of aerosol in Munich.

Our MAX-DOAS retrievals were also compared to independent data-sets: we used in-situ data from an ambient monitoring station for the intercomparison of surface aerosol extinction coefficients and MAX-DOAS derived $NO_2$ mixing ratios. A Pearson correlation coefficient of $R = 0.91$ was found between MAX-DOAS and in-situ measurements of surface level $NO_2$, however, the MAX-DOAS reports ∼50 % lower $NO_2$ mixing ratios. Lower $NO_2$ values measured by the MAX-DOAS are due





to the differences in spatial averaging. MAX-DOAS measurement of AOD was compared to AERONET data. The annual cycle was coherent with the MAX-DOAS measurements, and shows higher values in summer and lower values in winter.

Finally, we use tropospheric vertical column densities (VCDs) of $NO_2$ and HCHO derived from MAX-DOAS measurements to validate OMI and TROPOMI satellite observations. Monthly averaged data show good correlation with each other. However,

satellite observations are on average 30 % lower than the MAX-DOAS measurements. Underestimation of $NO_2$ and HCHO columns are largely related to the coarse spatial resolution of a-priori profiles of the satellite retrieval. Using MAX-DOAS observations as a-priori in satellite retrievals greatly reduce the underestimation.

In summary, our results demonstrate a wide range of applications of MAX-DOAS measurements in a global frame work, but also for investigations of the air quality in metropolitan areas. An obvious advantage is that different atmospheric components

can be retrieved simultaneously. Thus, for the understanding of details of and reasons for the interactions between trace gases, aerosols and meteorological variables, MAX-DOAS measurements can provide a substantial contribution, however, only the combination of different observation techniques (e.g. Schäfer et al., 2012; Geiß et al., 2017) and city scale resolving models (Vlemmix et al., 2015; Maronga et al., 2019a, b) can ultimately resolve the open questions and lead to (political) regulations for the future design of urban environments to meet high air quality standards.

*Code and data availability.*   The $M^3$ profile retrieval algorithm and MAX-DOAS data used in this study are available on request from the corresponding author (ka.chan@dlr.de).

**Appendix A**

*Competing interests.*   The authors declare that they have no conflict of interest.

*Acknowledgements.*   The work described in this paper was jointly supported by the DFG Major Research Instrumentation Programme (Grant

No. INST 86/1499-1 FUGG), the European Union's Horizon 2020 research and innovation programme through the ACTRIS-2 transnational access programme (Grant No. 654109), the Federal Minister of Transport and Digital Infrastructure (BMVI) through the framework of mFund (Grant No. 19F2065) and the German Aerospace Center (DLR) programmatic.





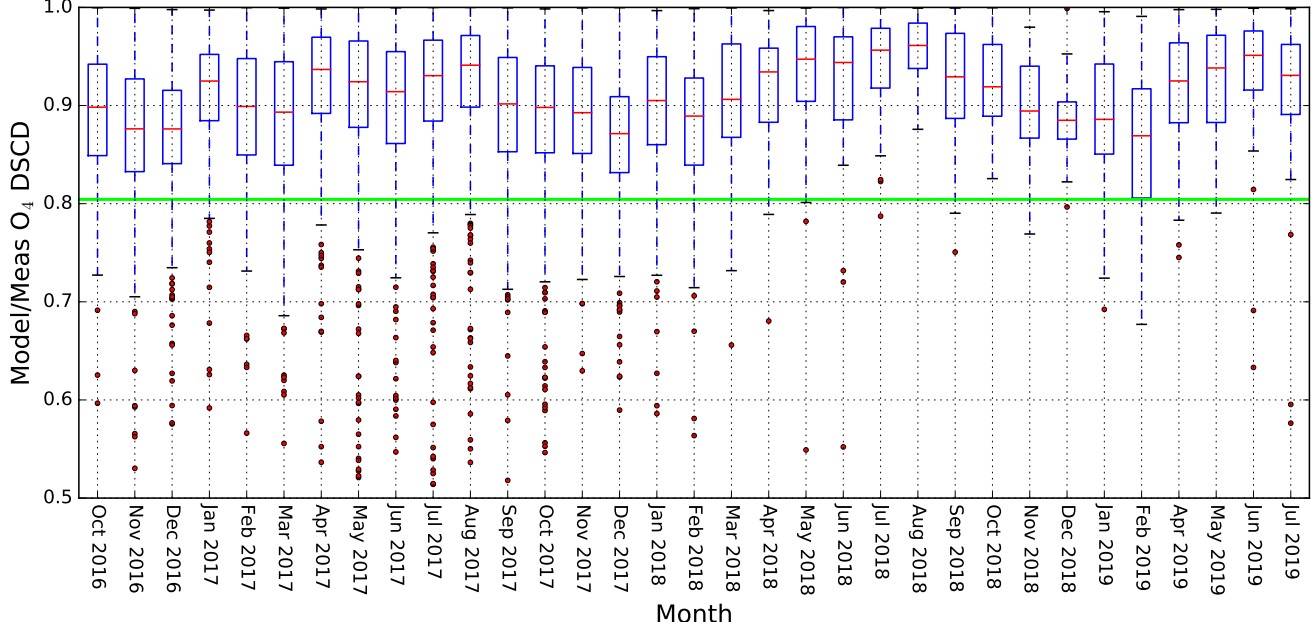

**Figure A1.** Monthly statistic of measured $O_4$ DSCD exceeding pure Rayleigh simulation. The ratios between modeled and measured $O_4$ DSCDs for observations taken with elevation angle of $15°$ are shown. The green line indicates the $10^{th}$ percentile (0.804).

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
