# Peer review of "MAX-DOAS measurements of tropospheric NO2 and HCHO in Munich and the comparison to OMI and TROPOMI satellite observations"

_Atmospheric Measurement Techniques, 2020_

## Referee Comment (RC1) · Anonymous Referee #1 · 4 May 2020

**General comments:**

The manuscript by Chan et al. presents a comparison work for satellite-based and ground-based $NO_2$ and HCHO measured in Munich. The work also evaluated the horizontal distributions of $NO_2$ and HCHO measured with different azimuth angles. The comparison process is accurate and comprehensive. Some of the findings are important and valuable to the research community. For example, using MAX-DOAS $NO_2$ profiles as a priori, the author recomputed OMI and TROPOMI $NO_2$ VCDs. This quantified influence of a priori $NO_2$ profiles in the satellite retrieval is interesting (i.e., the low-spatial-resolution a priori in original satellite data vs. MAX-DOAS derived a priori). The manuscript is well-written and should be published after addressing the following comments.

**Specific comments:**

P5 L12 to P6 L2. I think the $O_4$ scaling factor is still an interesting open question to the DOAS community. I am not challenging the validity of the $O_4$ scaling factor in this work (i.e., should or should not use $O_4$ scaling), but I feel the author's description is a bit misleading. I.e., one should at least mention those works (including Spinei et al., 2015; Wagner et al., 2019) that did not find it necessary to apply a scaling factor to bring model simulations and measurements into an agreement.

P7 L23-24. Please provide a quantitative description of the small effect of the radiative transfer simulation of $O_4$.

P10 L18. I think for this research work, a localized pixel-averaging map from TROPOMI is more useful than the map over Germany. For example, $NO_2$/HCHO map over Munich and surroundings might show more details of distribution features, i.e., whether there are any $NO_2$/HCHO hotspots near the MAX-DOAS site.

P11 L1-3 and L11-12. Without a good local map (masked with TROPOMI $NO_2$/HCHO), it is difficult for the reader to understand where are these emission sources (or hot spots), relative to the observation site. One should consider plot TROPOMI $NO_2$/HCHO (annual mean) masked over a map similar to Figure 1 (should be larger than Fig. 1, e.g., 50 km × 50 km). Also, proper labels (larger) for the discussed sources should be included, i.e., it is impossible to find where is the "English Garden", or "natural gas power plant" on Figure 1.

P12 L1-2. Since the y-axis for the four panels in Fig. 4 is very different, I am not sure the argument here is valid, i.e., the HCHO peak in the south and south-west during summer is less pronounced. The

absolute values from these two directions are about twice the corresponding values in the winter. Anyway, my point is the background level HCHO is different from winter to summer. Thus, to reveal the spatial distribution changes, one may needs to remove the background signal (e.g., mean HCHO or 5th to 10th percentile HCHO for each season). Also, given the very large error bars (1 std of HCHO), even after removing the background signal, I am not sure we can say the spatial variations from winter to summer is statistically significant.

P13 L6-8. I fully agree with the author that the biogenic emission from plants contributed to most of the signals shown in Fig. 5. But, is this possible to further separate the sources by divide the data into summer and winter periods? I guess in the winter HCHO dataset, one may see a better day of week variability. Any comments?

P13 L17-18. Please provide the calculated aerosol extinction to $NO_2$ ratios.

P14 L2. Please provide the calculated aerosol extinction to HCHO ratios.

P14 L9. Which model is used in the comparison? Please clarify.

P15 L2-4. Is the $\varepsilon_{surf}$ have any horizontal distribution pattern? For example, for the 180 degrees measurements, do we have larger $\varepsilon_{surf}$ than other directions (similar to the higher signal of $NO_2$ and HCHO from this azimuth angle)? For example, in Fig. 7b, do you have better/worse correlations for some directions?

P15 L22-29. I agree with the author that the sampling height could be one of the major reasons for this large systematic difference (50 %). If the author's hypothesis is correct, i.e., the difference is due to $NO_2$ vertical dispersion, one may see the systematic differences in different atmospheric conditions. For example, data collected around warm local noon (better vertical mixing) should show better agreement between MAX-DOAS surface $NO_2$ and in-situ $NO_2$, and vice versa. Any comments?

P18 L14-15. It is very nice to see the improvement from TROPOMI $NO_2$ when using MAX-DOAS derived profile as a priori. TM-5 is too coarse and high-spatial-resolution a priori is needed to capture enhanced local $NO_2$ signal. For North America, an hourly regional air quality forecasting model is used to re-calculate TROPOMI AMF (Griffin et al., 2019). For Europe, hourly CAMS regional model profiles available

at 0.1° resolution will be used in future TROPOMI data (e.g., Zhao et al., 2020). In general, I think these results found in current work look good. But, can the author give some comments on why there is an overestimate from the "OMI corr" point for February 2017?

**Technical corrections:**

P4 L9: Move the definition of DSCD to here.

P4 L9: Move the full name of $O_4$ (oxygen collision complex) to here.

P8 L10: Define $\Delta SCD_{ij}$, $\Delta SCD_{zenithj}$, and $\Delta z_j$.

Figs. 7b, 7d, and 8b. If these are colour coded density plots, please include proper colour bars.

Griffin, D., Zhao, X., McLinden, C. A., Boersma, K. F., Bourassa, A., Dammers, E., Degenstein, D., A., Eskes, H., Fehr, L., Fioletov, V., Hayden, K. L., Kharol, S. K., Li, S.-M., Makar, P., Martin, R. V., Mihele, C., Mittermeier, R. L., Krotkov, N., Sneep, M., Lamsal, L. N., ter Linden, M., van Geffen, J., Veefkind, P. and Wolde, M.: High resolution mapping of nitrogen dioxide with TROPOMI: First results and validation over the Canadian oil sands, Geophys. Res. Lett., 46(2), 1049–1060, doi:10.1029/2018GL081095, 2019.

Spinei, E., Cede, A., Herman, J., Mount, G. H., Eloranta, E., Morley, B., Baidar, S., Dix, B., Ortega, I., Koenig, T. and Volkamer, R.: Ground-based direct-sun DOAS and airborne MAX-DOAS measurements of the collision-induced oxygen complex, $O_2O_2$, absorption with significant pressure and temperature differences, Atmos. Meas. Tech., 8(2), 793–809, doi:10.5194/amt-8-793-2015, 2015.

Wagner, T., Beirle, S., Benavent, N., Bösch, T., Chan, K. L., Donner, S., Dörner, S., Fayt, C., Frieß, U., García-Nieto, D., Gielen, C., González-Bartolome, D., Gomez, L., Hendrick, F., Henzing, B., Jin, J. L., Lampel, J., Ma, J., Mies, K., Navarro, M., Peters, E., Pinardi, G., Puentedura, O., Puķīte, J., Remmers, J., Richter, A., Saiz-Lopez, A., Shaiganfar, R., Sihler, H., Roozendael, M. V., Wang, Y. and Yela, M.: Is a scaling factor required to obtain closure between measured and modelled atmospheric $O_4$ absorptions? An assessment of uncertainties of measurements and radiative transfer simulations for 2 selected days during the MAD-CAT campaign, Atmos. Meas. Tech., 12(5), 2745–2817, doi:10.5194/amt-12-2745-2019, 2019.

Zhao, X., Griffin, D., Fioletov, V., McLinden, C., Cede, A., Tiefengraber, M., Müller, M., Bognar, K., Strong, K., Boersma, F., Eskes, H., Davies, J., Ogyu, A. and Lee, S. C.: Assessment of the quality of TROPOMI high-spatial-resolution $NO_2$ data products in the Greater Toronto Area, Atmos. Meas. Tech., 13(4), 2131–2159, doi:10.5194/amt-13-2131-2020, 2020.

---

## Referee Comment (RC2) · Anonymous Referee #2 · 26 May 2020

The paper is about the two-dimensionally (2D) scanning Multi-AXis Differential Optical Absorption Spectroscopy (MAX-DOAS) observations of nitrogen dioxide (NO2) and formaldehyde (HCHO) in Munich. Vertical columns and vertical distribution profiles of aerosol extinction coefficient, NO2 and HCHO are retrieved from the 2D MAX-DOAS observations. The retrieved surface aerosol extinction coefficients and NO2 mixing ratios are compared to in situ monitoring data. The Pearson correlation coefficient (R) of surface NO2 mixing ratios and in situ monitoring data is 0.91. The aerosols optical depths (AODs) show good agreement as well (R=0.80) when compared to sun-photometer measurements. Following these results the tropospheric vertical column densities (VCDs) of NO2 and HCHO derived from the MAX-DOAS measurements are

used to validate OMI and TROPOMI satellite observations. Monthly averaged data show high correlations. However, satellite observations are on average 30% lower than the MAX-DOAS measurements. Furthermore, the MAX-DOAS observations are used to investigate the spatio-temporal characteristic of NO2 and HCHO in Munich. Analysis of the relations among aerosol, NO2 and HCHO shows higher aerosol to HCHO ratios in winter and a longer atmospheric lifetime of aerosol and HCHO is concluded. It is suggested from this analysis that secondary aerosol formation is the major source of aerosols in Munich. General comments MAX-DOAS observations are one of the measurements methods to detect the carcinogenic atmospheric pollutant HCHO which is originated by a lot of sources. Also, satellite observations of HCHO are available so that MAX-DOAS is an ideal ground-truthing method which should be applied for this task worldwide. The paper addresses relevant scientific questions within the scope of AMT. It completes the knowledge about NO2 and HCHO concentrations in urban area. The paper presents novel concepts, ideas and tools. The scientific methods and assumptions are valid and clearly outlined so that substantial conclusions are reached. The description of experiments and calculations are sufficiently complete and precise to allow their reproduction by fellow scientists. The quality and information of the figures is fine. The related work is well cited as well as the number and quality of references appropriate i.e. the authors give proper credit to related work and clearly indicate their own new/original contribution. The title and the abstract clearly reflects the contents of the paper. The overall presentation is well structured and clear. The language is fluent and precise. The mathematical formulae, symbols, abbreviations, and units are generally correctly defined. Specific Comments Please include at page 3, line 6 the name of the air quality monitoring station and a characterization of this station so that one can follow the analyses. Technical corrections Page 24, line : delete a dot.

---

## Author Comment (AC1) · 22 Jun 2020

Response to reviewer #1

We thank reviewer #1 for the time to carefully reading the manuscript and providing useful comments. We understand that these comments are positive on the scientific content of the manuscript while appropriate revisions and clarifications are necessary. We have addressed the reviewer's comments on a point to point basis as below for consideration. All page and line numbers refer to the marked-up version of the manuscript.

General comments:

[Figure]

The manuscript by Chan et al. presents a comparison work for satellite-based and ground-based NO2 and HCHO measured in Munich. The work also evaluated the horizontal distributions of NO2 and HCHO measured with different azimuth angles. The comparison process is accurate and comprehensive. Some of the findings are important and valuable to the research community. For example, using MAX-DOAS NO2 profiles as a priori, the author recomputed OMI and TROPOMI NO2 VCDs. This quantified influence of a priori NO2profilesin the satellite retrieval is interesting (i.e., the low-spatial-resolution a priori in original satellite data vs. MAX-DOAS derived a priori). The manuscript is well-written and should be published after addressing the following comments.

Specific comments:

P5 L12 to P6 L2. I think the O4 scaling factor is still an interesting open question to the DOAS community. I am not challenging the validity of the O4 scaling factor in this work (i.e., should or should not use O4 scaling), but I feel the author's description is a bit misleading. I.e., one should at least mention those works (including Spinei et al., 2015; Wagner et al., 2019) that did not find it necessary to apply a scaling factor to bring model simulations and measurements into an agreement.

Response: We followed the reviewer's comment and added the references and descriptions to studies which do not require any correction to bring observation and simulation together (page 4, line 12-13).

P7 L23-24. Please provide a quantitative description of the small effect of the radiative transfer simulation of O4.

Response: We have added the quantitative value for the surface albedo effect on the simulation of O4 DSCDs (page 7, line 29 to page 8, line 2).

P10 L18. I think for this research work, a localized pixel-averaging map from TROPOMI is more useful than the map over Germany. For example, NO2/HCHO map over Munich

and surroundings might show more details of distribution features, i.e., whether there areany NO2/HCHO hotspots near the MAX-DOAS site.

Response: In addition to the spatial distribution maps of NO2 and HCHO over Germany, we have also supplemented zoomed in maps of Munich and its surrounding areas. Major hotspots, e.g., power plant and airports, also marked on the maps (see figure 3).

P11 L1-3 and L11-12. Without a good local map (masked with TROPOMI NO2/HCHO), it is difficult for the reader to understand where are these emission sources (or hot spots), relative to the observation site. One should consider plot TROPOMI NO2/HCHO (annual mean) masked over a map similar to Figure 1(should be larger than Fig. 1, e.g., 50 km ×50 km). Also, proper labels (larger) for the discussed sources should be included, i.e., it is impossible to find where is the "English Garden", or "natural gas power plant" on Figure 1.

Response: See the response above. In addition, a city map is also included (figure 3e).

P12 L1-2. Since the y-axis for the four panels in Fig. 4 is very different, I am not sure the argument here is valid, i.e., the HCHO peak in the south and south-west during summer is less pronounced. The absolute values from these two directions are about twice the corresponding values in the winter. Anyway, my point is the background level HCHO is different from winter to summer. Thus, to reveal the spatial distribution changes, one may needs to remove the background signal (e.g., mean HCHO or 5th to 10th percentile HCHO for each season). Also, given the very large error bars (1 std of HCHO), even after removing the background signal, I am not sure we can say the spatial variations from winter to summer is statistically significant.

Response: The original idea of using separated plots for summer and winter time data is to show there is a big difference of the background value between summer and winter. We now followed the reviewer's comment and show normalized plots for

measurements at different azimuth angles by dividing the mean value.

Regarding to the comment of large error bars, the error bars are the $1\sigma$ standard deviation which represents the natural variation of the measurements, e.g., diurnal variation. These variations will not decrease even if we average large amounts of data, while the errors of the measurement values are very small as it is an average of a large number of data. As the errors are too small to be visible in the plots, therefore, we decided to show the $1\sigma$ standard deviation instead. We have further clarified this point in the manuscript (page 12, line 1-5).

P13 L6-8. I fully agree with the author that the biogenic emission from plants contributed to most of the signals shown in Fig. 5. But, is this possible to further separate the sources by divide the data into summer and winter periods? I guess in the winter HCHO dataset, one may see a better day of week variability. Any comments?

Response: We followed the reviewer's comment and separated the day of week analysis into winter and summer periods. For NO2, a more significant weekend reduction can be observed in summer, which is due to shorter atmospheric lifetime and less accumulation from weekdays. For HCHO, the weekly pattern is much less pronounced during winter (no weekend reduction can be observed). In winter, HCHO levels observed on Sunday are even slightly higher than that of the weekday average. The anthropogenic sources of HCHO in the troposphere include the oxidation of various long lifetime VOCs, such as, methane. Their lifetimes are even longer in winter and therefore result in a less significant weekly pattern. This information is included in the revised manuscript (page 13, line 13-30).

P13 L17-18. Please provide the calculated aerosol extinction to NO2 ratios.

Response: We have supplemented the aerosol extinction to NO2 ratios for both summer and winter in the manuscript (page 14, line 7 to page 15, line 1).

P14 L2. Please provide the calculated aerosol extinction to HCHO ratios.

Response: We have supplemented the aerosol extinction to HCHO ratios for both summer and winter in the manuscript (page 15, line 3-4).

P14 L9. Which model is used in the comparison? Please clarify.

Response: The model refers to the multiple linear regression model. We have revised the sentence to avoid confusion (page 15, line 10).

P15 L2-4. Is the $\varepsilon$surf have any horizontal distribution pattern? For example, for the 180 degrees measurements, do we have larger $\varepsilon$surf than other directions (similar to the higher signal of NO2 and HCHO from this azimuth angle)? For example, in Fig. 7b, do you have better/worse correlations for some directions?

Response: As the in-situ monitor station is located northwest of the MAX-DOAS measurement site, MAX-DOAS measurements of aerosol extinction at surface layer with azimuth angle of $315°$ agree the best with the in-situ data with a correlation coefficient of 0.82. The result indicates the strong spatial variation of aerosols in Munich and a single in-situ monitor is not representative for the general pollution condition in the city. We have supplemented a corresponding statement in the manuscript (page 16, line 19-22).

P15 L22-29. I agree with the author that the sampling height could be one of the major reasons for this large systematic difference (50 %). If the author's hypothesis is correct, i.e., the difference is due to NO2 vertical dispersion, one may see the systematic differences in different atmospheric conditions. For example, data collected around warm local noon (better vertical mixing) should show better agreement between MAX-DOAS surface NO2 and in-situ NO2, and vice versa. Any comments?

Response: Following the reviewer's comment, we have separated the measurements into few categories by meteorological factors, such as, temperature and wind speed, for analysis. However, we do not see any significant improvement of the agreement between the MAX-DOAS and in-situ measurements. In addition, we have linearly extrapolated the MAX-DOAS measurements to near street level (15m a.g.l.) using the lowest two layers of the NO2 vertical profile retrieval. The extrapolated near street level NO2 concentrations are on average only ~10% higher. However, the discrepancy between MAX-DOAS and in-situ measurements remains quite large. The result indicates a stronger enhancement of NO2 level at near street level compared to the upper part of the mixing layer. The vertical mixing of pollutants in an urban environment is rather complicated. The atmospheric processes are especially complicated in the lowest several tens of meters where pollutants emitted from tail pipes are dispersed to the ambient environment. These processes are strongly dependent on many factors, such as, the urban street configurations, emission characteristics and meteorological factors. Higher spatio-temporal resolution measurements and a proper CFD model are required to better investigate the pollution dispersion effect in urban environment. However, this topic is beyond the scope of this study. A more detailed description and explanation is included in the manuscript (page 16, line 34 to page 17, line 3).

P18 L14-15. It is very nice to see the improvement from TROPOMI NO2 when using MAX-DOAS derived profile as a priori. TM-5 is too coarse and high-spatial-resolution a priori is needed to capture enhanced local NO2 signal. For North America, an hourly regional air quality forecasting model is used to recalculate TROPOMI AMF (Griffin et al., 2019). For Europe, hourly CAMS regional model profiles available at 0.1° resolution will be used in future TROPOMI data (e.g., Zhao et al., 2020). In general, I think these results found in current work look good. But, can the author give some comments on why there is an overestimate from the "OMI corr" point for February 2017?

Response: We have supplemented the references to the recent relevant studies (page 20, line 13-14). For the OMI measurement on Feb 2017 exceeding the MAX-DOAS value, this is mainly because there are only three valid OMI measurements during the month due to cloudiness and row anomaly issue while the MAX-DOAS has 20 valid measurements in Feb 2017. We have supplemented this explanation in the manuscript (page 19, line 14-15).

Technical corrections:

P4 L9: Move the definition of DSCD to here.

Response: Done.

P4 L9: Move the full name of O4 (oxygen collision complex) to here.

Response: Done.

P8 L10: Define $\Delta SCD_{ij}$, $\Delta SCD$ zenith$_j$, and $\Delta z_j$. Figs.

Response: Done.

7b, 7d, and 8b. If these are colour coded density plots, please include proper colour bars.

Response: Done.

Griffin, D., Zhao, X., McLinden, C. A., Boersma, K. F., Bourassa, A., Dammers, E., Degenstein, D., A., Eskes, H., Fehr, L., Fioletov, V., Hayden, K. L., Kharol, S. K., Li, S.-M., Makar, P., Martin, R. V., Mihele, C., Mittermeier, R. L., Krotkov, N., Sneep, M., Lamsal, L. N., terLinden, M., van Geffen, J., Veefkind, P. and Wolde, M.: High resolution mapping of nitrogen dioxide with TROPOMI: First results and validation over the Canadian oil sands, Geophys. Res. Lett., 46(2), 1049–1060, doi:10.1029/2018GL081095, 2019.

Spinei, E., Cede, A., Herman, J., Mount, G. H., Eloranta, E., Morley, B., Baidar, S., Dix, B., Ortega, I., Koenig, T. and Volkamer, R.: Ground-based direct-sun DOAS and airborne MAX-DOAS measurements of the collision-induced oxygen complex, O2O2, absorption with significant pressure and temperature differences, Atmos. Meas. Tech., 8(2), 793–809, doi:10.5194/amt-8-793-2015, 2015.

Wagner, T., Beirle, S., Benavent, N., Bösch, T., Chan, K. L., Donner, S., Dörner, S., Fayt, C., Frieß, U., García-Nieto, D., Gielen, C., González-Bartolome, D., Gomez, L., Hendrick, F., Henzing, B., Jin, J. L., Lampel, J., Ma, J., Mies, K., Navarro, M., Peters,

E., Pinardi, G., Puentedura, O., PuÄůÄńte, J., Remmers, J., Richter, A., Saiz-Lopez, A., Shaiganfar, R., Sihler, H., Roozendael, M. V., Wang, Y. and Yela, M.: Is a scaling factor required to obtain closure between measured and modelled atmospheric O4 absorptions? An assessment of uncertainties of measurements and radiative transfer simulations for 2 selected days during the MAD-CAT campaign, Atmos. Meas. Tech., 12(5), 2745–2817, doi:10.5194/amt-12-2745-2019, 2019.

Zhao, X., Griffin, D., Fioletov, V., McLinden, C., Cede, A., Tiefengraber, M., Müller, M., Bognar, K., Strong, K., Boersma, F., Eskes, H., Davies, J., Ogyu, A. and Lee, S. C.: Assessment of the quality of TROPOMI high-spatial-resolution NO2 data products in the Greater Toronto Area, Atmos. Meas. Tech., 13(4), 2131–2159, doi:10.5194/amt-13-2131-2020, 2020.

---

## Author Comment (AC2) · 22 Jun 2020

Response to reviewer #2

We thank reviewer #2 for the useful comments. We understand that these comments are mostly positive while minor corrections are necessary. We have addressed the reviewer's comments on a point to point basis as below for consideration. All page and line numbers refer to the marked-up version of the manuscript.

The paper is about the two-dimensionally (2D) scanning Multi-AXis Differential Optical Absorption Spectroscopy (MAX-DOAS) observations of nitrogen dioxide (NO2) and

formaldehyde (HCHO) in Munich. Vertical columns and vertical distribution profiles of aerosol extinction coefficient, NO2 and HCHO are retrieved from the 2D MAX-DOAS observations. The retrieved surface aerosol extinction coefficients and NO2 mixing ratios are compared to in situ monitoring data. The Pearson correlation coefficient (R) of surface NO2 mixing ratios and in situ monitoring data is 0.91. The aerosols optical depths (AODs) show good agreement as well (R=0.80) when compared to sun-photometer measurements. Following these results the tropospheric vertical column densities (VCDs) of NO2 and HCHO derived from the MAX-DOAS measurements are used to validate OMI and TROPOMI satellite observations. Monthly averaged data show high correlations. However, satellite observations are on average 30% lower than the MAX-DOAS measurements. Furthermore, the MAX-DOAS observations are used to investigate the spatio-temporal characteristic of NO2 and HCHO in Munich. Analysis of the relations among aerosol, NO2 and HCHO shows higher aerosol to HCHO ratios in winter and a longer atmospheric lifetime of aerosol and HCHO is concluded. It is suggested from this analysis that secondary aerosol formation is the major source of aerosols in Munich.

General comments

MAX-DOAS observations are one of the measurements methods to detect the carcino-genic atmospheric pollutant HCHO which is originated by a lot of sources. Also, satel-lite observations of HCHO are available so that MAX-DOAS is an ideal ground-truthing method which should be applied for this task worldwide. The paper addresses relevant scientific questions within the scope of AMT. It completes the knowledge about NO2 and HCHO concentrations in urban area. The paper presents novel concepts, ideas and tools. The scientific methods and assumptions are valid and clearly outlined so that substantial conclusions are reached. The description of experiments and calculations are sufficiently complete and precise to allow their reproduction by fellow scientists. The quality and information of the figures is fine. The related work is well cited as well as the number and quality of references appropriate i.e. the authors give proper credit
to related work and clearly indicate their own new/original contribution. The title and the abstract clearly reflects the contents of the paper. The overall presentation is well structured and clear. The language is fluent and precise. The mathematical formulae, symbols, abbreviations, and units are generally correctly defined.

Specific Comments

Please include at page 3, line 6 the name of the air quality monitoring station and a characterization of this station so that one can follow the analyses.

Response: We have supplemented the name, the coordinate and the characteristic of the air quality monitor station in the manuscript (page 3, line 8-9).

Technical corrections

Page 24, line: delete a dot.

Response: Done.

---

## Author Response (AR2)

Response to editor

We thank the editor for the useful comments. We have addressed the editor's comments on a point to point basis as below for consideration. All page and line numbers refer to the marked-up version of the manuscript.

Comments to the Author:

Overall, the reviews and revisions were good and I find this manuscript acceptable for publication. The revised text has some small issues that should be resolved before publication.

page 1, line 9. The sentence "Analysis of the relations among aerosol, NO2 and HCHO shows higher aerosol to HCHO ratios in winter indicating a longer atmospheric lifetime of aerosol and HCHO." is a bit confusing. The ratio of aerosol to HCHO is mentioned but then both aerosol and HCHO are said to have a longer lifetime. I think that page 15, lines 12-14 really argue that the larger ratio of aerosol / HCHO in winter is a result of longer HCHO lifetime in winter than summer. Please clarify this with respect to the manuscript's arguments.

Response: We have revised the sentence to "Analysis of the relations between aerosol, NO2 and HCHO shows higher aerosol to HCHO ratios in winter which reflects a longer atmospheric lifetime of secondary aerosol and HCHO during winter." (page 1, line 11-13).

page 11, Figure 3 panels c and d are a bit incongruous with the text because that text says that HCHO is homogeneous, but panel d looks like there is significant structure. Please change panels c and d to have the lower limit be zero. I would also note that the panel c has a different scale than panel a in the caption. I would also change the end of the caption to say "...taken from Google maps (https://www.google.com/maps/)." -- capitalizing the company "Google" and making maps plural.

Response: As the color range of figure 3d is much narrower than that of figure 3b, the HCHO structures shown in figure 3d are indeed not very significant. Nevertheless, we followed the editor comment and revised the description in the text (page 11, line 3-4). In addition, we have changed the color scale of figure 3c and d, so that they are consistence with figure 3a and b. We have also capitalizing the company name "Google" and making maps plural. Response: Done.

page 12, line 5, say "... better horizontal mixing of NO2..." because the shallower layer is caused by less vertical mixing of NO2.

Response: Done.

We thank reviewer #1 for the time to carefully reading the manuscript and providing useful comments. We understand that these comments are positive on the scientific content of the manuscript while appropriate revisions and clarifications are necessary. We have addressed the reviewer's comments on a point to point basis as below for consideration. All page and line numbers refer to the marked-up version of the manuscript.

General comments:

The manuscript by Chan et al. presents a comparison work for satellite-based and ground-based NO2 and HCHO measured in Munich. The work also evaluated the horizontal distributions of NO2 and HCHO measured with different azimuth angles. The comparison process is accurate and comprehensive. Some of the findings are important and valuable to the research community. For example, using MAX-DOAS NO2 profiles as a priori, the author recomputed OMI and TROPOMI NO2 VCDs. This quantified influence of a priori NO2profilesin the satellite retrieval is interesting (i.e., the low-spatial-resolution a priori in original satellite data vs. MAX-DOAS derived a priori). The manuscript is well-written and should be published after addressing the following comments.

Specific comments:

P5 L12 to P6 L2. I think the O4 scaling factor is still an interesting open question to the DOAS community. I am not challenging the validity of the O4 scaling factor in this work (i.e., should or should not use O4 scaling), but I feel the author's description is a bit misleading. I.e., one should at least mention those works (including Spinei et al., 2015; Wagner et al., 2019) that did not find it necessary to apply a scaling factor to bring model simulations and measurements into an agreement.

Response: We followed the reviewer's comment and added the references and descriptions to studies which do not require any correction to bring observation and simulation together (page 4, line 12-13).

P7 L23-24. Please provide a quantitative description of the small effect of the radiative transfer simulation of O4.

Response: We have added the quantitative value for the surface albedo effect on the simulation of O4 DSCDs (page 7, line 29 to page 8, line 2).

P10 L18. I think for this research work, a localized pixel-averaging map from TROPOMI is more useful than the map over Germany. For example, NO2/HCHO map over Munich and surroundings might show more details of distribution features, i.e., whether there areany NO2/HCHO hotspots near the MAX-DOAS site.

Response: In addition to the spatial distribution maps of NO2 and HCHO over Germany, we have also supplemented zoomed in maps of Munich and its surrounding areas. Major hotspots, e.g., power plant and airports, also marked on the maps (see figure 3).

P11 L1-3 and L11-12. Without a good local map (masked with TROPOMI NO2/HCHO), it is difficult for the reader to understand where are these emission sources (or hot spots), relative to the observation site. One should consider plot TROPOMI NO2/HCHO (annual mean) masked over a map similar to Figure 1(should be larger than Fig. 1, e.g., 50 km ×50 km). Also, proper labels (larger) for the discussed sources should be included, i.e., it is impossible to find where is the "English Garden", or "natural gas power plant" on Figure 1.

Response: See the response above. In addition, a city map is also included (figure 3e).

P12 L1-2. Since the y-axis for the four panels in Fig. 4 is very different, I am not sure the argument here is valid, i.e., the HCHO peak in the south and south-west during summer is less pronounced. The absolute values from these two directions are about twice the corresponding values in the winter. Anyway, my point is the background level HCHO is different from winter to summer. Thus, to reveal the spatial distribution changes, one may needs to remove the background signal (e.g., mean HCHO or $5^{th}$ to $10^{th}$ percentile HCHO for each season). Also, given the very large error bars (1 std of HCHO), even after removing the background signal, I am not sure we can say the spatial variations from winter to summer is statistically significant.

Response: The original idea of using separated plots for summer and winter time data is to show there is a big difference of the background value between summer and winter. We now followed the reviewer's comment and show normalized plots for measurements at different azimuth angles by dividing the mean value.

Regarding to the comment of large error bars, the error bars are the $1\sigma$ standard deviation which represents the natural variation of the measurements, e.g., diurnal variation. These variations will not decrease even if we average large amounts of data, while the errors of the measurement values are very small as it is an average of a large number of data. As the errors are too small to be visible in the

plots, therefore, we decided to show the 1σ standard deviation instead. We have further clarified this point in the manuscript (page 12, line 1-5).

P13 L6-8. I fully agree with the author that the biogenic emission from plants contributed to most of the signals shown in Fig. 5. But, is this possible to further separate the sources by divide the data into summer and winter periods? I guess in the winter HCHO dataset, one may see a better day of week variability. Any comments?

Response: We followed the reviewer's comment and separated the day of week analysis into winter and summer periods. For NO2, a more significant weekend reduction can be observed in summer, which is due to shorter atmospheric lifetime and less accumulation from weekdays. For HCHO, the weekly pattern is much less pronounced during winter (no weekend reduction can be observed). In winter, HCHO levels observed on Sunday are even slightly higher than that of the weekday average. The anthropogenic sources of HCHO in the troposphere include the oxidation of various long lifetime VOCs, such as, methane. Their lifetimes are even longer in winter and therefore result in a less significant weekly pattern. This information is included in the revised manuscript (page 13, line 13-30).

P13 L17-18. Please provide the calculated aerosol extinction to NO2 ratios.

Response: We have supplemented the aerosol extinction to NO2 ratios for both summer and winter in the manuscript (page 14, line 7 to page 15, line 1).

P14 L2. Please provide the calculated aerosol extinction to HCHO ratios.

Response: We have supplemented the aerosol extinction to HCHO ratios for both summer and winter in the manuscript (page 15, line 3-4).

P14 L9. Which model is used in the comparison? Please clarify.

Response: The model refers to the multiple linear regression model. We have revised the sentence to avoid confusion (page 15, line 10).

P15 L2-4. Is the εsurf have any horizontal distribution pattern? For example, for the 180 degrees measurements, do we have larger εsurf than other directions (similar to the higher signal of NO2 and

HCHO from this azimuth angle)? For example, in Fig. 7b, do you have better/worse correlations for some directions?

Response: As the in-situ monitor station is located northwest of the MAX-DOAS measurement site, MAX-DOAS measurements of aerosol extinction at surface layer with azimuth angle of 315° agree the best with the in-situ data with a correlation coefficient of 0.82. The result indicates the strong spatial variation of aerosols in Munich and a single in-situ monitor is not representative for the general pollution condition in the city. We have supplemented a corresponding statement in the manuscript (page 16, line 19-22).

P15 L22-29. I agree with the author that the sampling height could be one of the major reasons for this large systematic difference (50 %). If the author's hypothesis is correct, i.e., the difference is due to $NO_2$ vertical dispersion, one may see the systematic differences in different atmospheric conditions. For example, data collected around warm local noon (better vertical mixing) should show better agreement between MAX-DOAS surface $NO_2$ and in-situ $NO_2$, and vice versa. Any comments?

Response: Following the reviewer's comment, we have separated the measurements into few categories by meteorological factors, such as, temperature and wind speed, for analysis. However, we do not see any significant improvement of the agreement between the MAX-DOAS and in-situ measurements. In addition, we have linearly extrapolated the MAX-DOAS measurements to near street level (15m a.g.l.) using the lowest two layers of the $NO_2$ vertical profile retrieval. The extrapolated near street level $NO_2$ concentrations are on average only ~10% higher. However, the discrepancy between MAX-DOAS and in-situ measurements remains quite large. The result indicates a stronger enhancement of $NO_2$ level at near street level compared to the upper part of the mixing layer. The vertical mixing of pollutants in an urban environment is rather complicated. The atmospheric processes are especially complicated in the lowest several tens of meters where pollutants emitted from tail pipes are dispersed to the ambient environment. These processes are strongly dependent on many factors, such as, the urban street configurations, emission characteristics and meteorological factors. Higher spatio-temporal resolution measurements and a proper CFD model are required to better investigate the pollution dispersion effect in urban environment. However, this topic is beyond the scope of this study. A more detailed description and explanation is included in the manuscript (page 16, line 34 to page 17, line 3).

P18 L14-15. It is very nice to see the improvement from TROPOMI $NO_2$ when using MAX-DOAS derived profile as a priori. TM-5 is too coarse and high-spatial-resolution a priori is needed to capture enhanced local $NO_2$ signal. For North America, an hourly regional air quality forecasting model is used to recalculate TROPOMI AMF (Griffin et al., 2019). For Europe, hourly CAMS regional model profiles available at 0.1° resolution will be used in future TROPOMI data (e.g., Zhao et al., 2020). In general, I think these results found in current work look good. But, can the author give some comments on why there is an overestimate from the "OMI corr" point for February 2017?

Response: We have supplemented the references to the recent relevant studies (page 20, line 13-14). For the OMI measurement on Feb 2017 exceeding the MAX-DOAS value, this is mainly because there are only three valid OMI measurements during the month due to cloudiness and row anomaly issue while the MAX-DOAS has 20 valid measurements in Feb 2017. We have supplemented this explanation in the manuscript (page 19, line 14-15).

Technical corrections:

P4 L9: Move the definition of DSCD to here.

Response: Done.

P4 L9: Move the full name of O4 (oxygen collision complex) to here.

Response: Done.

P8 L10: Define $\Delta SCD_{ij}$, $\Delta SCD$ zenith$_j$, and $\Delta z_j$. Figs.

Response: Done.

7b, 7d, and 8b. If these are colour coded density plots, please include proper colour bars.

Response: Done.

Griffin, D., Zhao, X., McLinden, C. A., Boersma, K. F., Bourassa, A., Dammers, E., Degenstein, D., A., Eskes, H., Fehr, L., Fioletov, V., Hayden, K. L., Kharol, S. K., Li, S.-M., Makar, P., Martin, R. V., Mihele, C., Mittermeier, R. L., Krotkov, N., Sneep, M., Lamsal, L. N., terLinden, M., van Geffen, J., Veefkind, P. and Wolde, M.: High resolution mapping of nitrogen dioxide with TROPOMI: First results and validation over the Canadian oil sands, Geophys. Res. Lett., 46(2), 1049–1060, doi:10.1029/2018GL081095, 2019.

Spinei, E., Cede, A., Herman, J., Mount, G. H., Eloranta, E., Morley, B., Baidar, S., Dix, B., Ortega, I., Koenig, T. and Volkamer, R.: Ground-based direct-sun DOAS and airborne MAX-DOAS measurements of the collision-induced oxygen complex, O2O2, absorption with significant pressure and temperature differences, Atmos. Meas. Tech., 8(2), 793–809, doi:10.5194/amt-8-793-2015, 2015.

Wagner, T., Beirle, S., Benavent, N., Bösch, T., Chan, K. L., Donner, S., Dörner, S., Fayt, C., Frieß, U., García-Nieto, D., Gielen, C., González-Bartolome, D., Gomez, L., Hendrick, F., Henzing, B., Jin, J. L., Lampel, J., Ma, J., Mies, K., Navarro, M., Peters, E., Pinardi, G., Puentedura, O., Puķīte, J., Remmers, J., Richter, A., Saiz-Lopez, A., Shaiganfar, R., Sihler, H., Roozendael, M. V., Wang, Y. and Yela, M.: Is a scaling factor required to obtain closure between measured and modelled atmospheric O4 absorptions? An assessment of uncertainties of measurements and radiative transfer simulations for 2 selected days during the MAD-CAT campaign, Atmos. Meas. Tech., 12(5), 2745–2817, doi:10.5194/amt-12-2745-2019, 2019.

Zhao, X., Griffin, D., Fioletov, V., McLinden, C., Cede, A., Tiefengraber, M., Müller, M., Bognar, K., Strong, K., Boersma, F., Eskes, H., Davies, J., Ogyu, A. and Lee, S. C.: Assessment of the quality of TROPOMI high-spatial-resolution NO2 data products in the Greater Toronto Area, Atmos. Meas. Tech., 13(4), 2131–2159, doi:10.5194/amt-13-2131-2020, 2020.

Response to reviewer #2

We thank reviewer #2 for the useful comments. We understand that these comments are mostly positive while minor corrections are necessary. We have addressed the reviewer's comments on a point to point basis as below for consideration. All page and line numbers refer to the marked-up version of the manuscript.

The paper is about the two-dimensionally (2D) scanning Multi-AXis Differential Optical Absorption Spectroscopy (MAX-DOAS) observations of nitrogen dioxide (NO2) and formaldehyde (HCHO) in Munich. Vertical columns and vertical distribution profiles of aerosol extinction coefficient, NO2 and HCHO are retrieved from the 2D MAX-DOAS observations. The retrieved surface aerosol extinction coefficients and NO2 mixing ratios are compared to in situ monitoring data. The Pearson correlation coefficient (R) of surface NO2 mixing ratios and in situ monitoring data is 0.91. The aerosols optical depths (AODs) show good agreement as well (R=0.80) when compared to sun-photometer measurements. Following these results the tropospheric vertical column densities (VCDs) of NO2 and HCHO derived from the MAX-DOAS measurements are used to validate OMI and TROPOMI satellite observations. Monthly averaged data show high correlations. However, satellite observations are on average 30% lower than the MAX-DOAS measurements. Furthermore, the MAX-DOAS observations are used to investigate the spatio-temporal characteristic of NO2 and HCHO in Munich. Analysis of the relations among aerosol, NO2 and HCHO shows higher aerosol to HCHO ratios in winter and a longer atmospheric lifetime of aerosol and HCHO is concluded. It is suggested from this analysis that secondary aerosol formation is the major source of aerosols in Munich.

General comments

MAX-DOAS observations are one of the measurements methods to detect the carcinogenic atmospheric pollutant HCHO which is originated by a lot of sources. Also, satellite observations of HCHO are available so that MAX-DOAS is an ideal ground-truthing method which should be applied for this task worldwide. The paper addresses relevant scientific questions within the scope of AMT. It completes the knowledge about NO2 and HCHO concentrations in urban area. The paper presents novel concepts, ideas and tools. The scientific methods and assumptions are valid and clearly outlined so that substantial conclusions are reached. The description of experiments and calculations are sufficiently complete and precise to allow their reproduction by fellow scientists. The quality and information of the figures is fine. The related work is well cited as well as the number and quality of references appropriate i.e. the authors give proper credit to related work and clearly indicate their own new/original contribution. The title and the abstract clearly reflects the contents of the paper. The overall presentation is well structured and clear. The language is fluent and precise. The mathematical formulae, symbols, abbreviations, and units are generally correctly defined.

Specific Comments

Please include at page 3, line 6 the name of the air quality monitoring station and a characterization of this station so that one can follow the analyses.

Response: We have supplemented the name, the coordinate and the characteristic of the air quality monitor station in the manuscript (page 3, line 8-9).

Technical corrections

Page 24, line: delete a dot.

Response: Done.

[revised manuscript text omitted]